# Possible implications of enhanced CFC-11 concentrations on ozone

Martin Dameris, Patrick Jöckel, Matthias Nützel

Deutsches Zentrum für Luft- und Raumfahrt, Institut für Physik der Atmosphäre, Oberpfaffenhofen, Germany

*Correspondence to*: Martin Dameris (martin.dameris@dlr.de)

**Abstract.** This numerical model study is motivated by the observed global deviation from assumed emissions of chlorofluorocarbon-11 (CFC-11, $CFCl_3$) in recent years. Montzka et al. (2018) discussed a strong deviation of the assumed emissions of CFC-11 in the past 15 years, which indicates a violation of the Montreal Protocol for the protection of the ozone layer. An investigation is performed based on Chemistry-Climate Model (CCM) simulations that analyze the

consequences of an enhanced CFC-11 surface mixing ratio. In comparison to a reference simulation (REF-C2), where a decrease of the CFC-11 surface mixing ratio of about 50% is assumed from the early 2000s to the middle of the century (i.e. mixing ratio in full compliance with the Montreal Protocol agreement), two sensitivity simulations are carried out: One simulation in which after the year 2002 the CFC-11 surface mixing ratio is kept constant until 2050 (SEN-C2-fCFC11_2050); this allows a qualitative estimate of possible consequences of high-level stable CFC-11 surface mixing ratio

on the ozone layer. In the second sensitivity simulation, which is branched off from the first sensitivity simulation, it is assumed that starting in year 2020 the Montreal Protocol is fully implemented again, which leads to a delayed decrease of CFC-11 in this simulation (SEN-C2-fCFC11_2020) compared to the reference simulation; this enables a rough and most likely upper-limit assessment of how much the unexpected CFC-11 emissions to date have already affected ozone. In all three simulations climate evolves under the same greenhouse gas scenario (i.e. RCP 6.0) and all other ozone depleting

substances are declining according to this scenario. Differences between the reference (REF-C2) and the two sensitivity simulations (SEN-C2-fCFC11_2050 and SEN-C2-fCFC11_2020) are discussed. In the SEN-C2-fCFC11_2050 simulation the total column ozone (TCO) in the 2040s (i.e. the years 2041-2050) is particularly affected in both polar regions in winter and spring. Maximum discrepancies of TCO are identified with reduced ozone values of up to around 30 Dobson Units in the Southern Hemisphere (SH) polar region during SH spring (in the order of 15%). An analysis of the respective partial

column ozone (PCO) for the stratosphere indicates that strongest ozone changes are calculated for the polar lower stratosphere, where they are mainly driven by the enhanced stratospheric chlorine content and associated heterogeneous chemical processes. Furthermore, it turns out that the calculated ozone changes, especially in the upper stratosphere, are surprisingly small. For the first time in such a scenario we perform a complete ozone budget analysis regarding the production and loss cycles. The budget analysis shows that in the upper stratosphere the additional ozone depletion due to

the catalysis by reactive chlorine is compensated partly by other processes related to enhanced ozone production or reduced ozone loss, for instance from nitrous oxide ($NO_x$). Based on the analysis of the SEN-C2-fCFC11_2020 simulation it turned

out that no major ozone changes can be expected after year 2050, which are related to the enhanced CFC-11 emissions in recent years.

## 1 Introduction

The estimation of the future evolution of the ozone layer is a central part of the UNEP/WMO Scientific Assessment of
Ozone Depletion. For that reason Chemistry-Climate Models (CCM) are carrying out long-term simulations (for several decades). These models are performing comprehensive numerical simulations under well-defined boundary conditions which are prescribing possible future changes of ozone depleting substances (ODSs), particularly those related to changes of chlorofluorocarbon (CFC) concentrations. In recent years, model guidelines have been defined, to facilitate the inter-comparison of CCM results from different modelling groups world-wide. For instance, in 2012 the Chemistry-Climate
Model Initiative (CCMI), under the umbrella of IGAC/SPARC, defined the boundary conditions for the next round of coordinated reference (REF) and sensitivity (SEN) simulations (Eyring et al., 2013). The boundary conditions for the CCM simulations consider not only the expected changes of ODSs according to the regulations of the Montreal Protocol and its amendments, but also the influence of different climate change scenarios. Here, the greenhouse gas concentrations for the Representative Concentration Pathways (RCPs) adopted by the IPCC for its 5[th] Assessment Report (AR5) in 2014 (van
Vuuren et al., 2011) were recommended. The suggested reference simulation for the future (REF-C2) assumes full compliance with the Montreal Protocol, expecting more or less no further production of CFCs.

More recently the respective CCM results were presented and discussed in several scientific papers, for instance by Dhomse et al. (2018). Among other things the question of ozone recovery was investigated. Further, it was analyzed how the speed of recovery and the return date are affected by the expected decrease of ODSs and by climate change. The results of the CCM
simulations were taken into account as the foundation for the latest WMO ozone report (WMO, 2018).

During the preparation phase of WMO (2018) a paper by Montzka et al. (2018) was published, indicating a clear deviation of the expected surface concentration of chlorofluorocarbon-11 (CFC-11, $CFCl_3$) in the past 15 years. Observational datasets discussed by Montzka et al. (2018; M18) showed that (1) in the last 10 years (until 2017) the decline of CFC-11 surface mixing ratios was obviously much slower than expected (see Figure 1a in M18); (2) the decline of CFC-11 surface mixing
ratios was nearly constant from 2002 to 2012, whereas in the following years the decrease of CFC-11 surface mixing ratios decelerated (here until 2017; see Figure 1b in M18). The measurements indicated increased CFC-11 emissions after 2012 (see Figure 2a in M18; see also Figure ES-2 in WMO, 2018). Montzka and colleagues (M18) mentioned that these observations imply "a gap in our understanding of CFC-11 sources and sinks since the early 2000s".

Based on these findings a significant impact on the recovery of the ozone layer seems to be possible, in particular, if CFC-11
emissions do not decline as previously anticipated (e.g. Daniel and Velders et al., 2011; Carpenter and Reimann et al., 2014). Therefore the assumption of decreasing CFC-11 surface mixing ratio in the future by CCMI is partly questionable. Currently

the future evolution of CFC-11 emissions is uncertain (Harris et al., 2019). Therefore, for this numerical study we cannot decide, which future scenario is most likely and also the possible ranges of future CFC-11 emission changes are difficult to estimate. Our approach is to employ a high emission scenario of CFC-11 in a sensitivity simulation (SEN-C2-fCFC11_2050) by imposing constant surface mixing ratios of CFC-11 from 2002 onwards (Figure 1). The sensitivity simulation covers the period from 2002 to 2050. This sensitivity simulation should not be considered as a specific future scenario that we deem likely. As said this model set-up can be taken as a high-level scenario regarding CFC-11 background conditions. The reason for starting this sensitivity simulation in the year 2002 is motivated by the findings of M18 showing that from that year on the observed emissions started to diverge from the expected emissions of CFC-11 (e.g. Figure 2a in M18). We note that stable CFC-11 emissions are not equal to stable CFC-11 surface concentrations. But due to the fact that the future evolution of CFC-11 emissions (and also the surface mixing ratio) is uncertain, such a simplified (crude) assumption of a constant surface mixing ratio is absolutely justified as a high-level scenario dating back to the point when expected emissions and observations started to diverge. An alternative sensitivity simulation could be created assuming a later starting date (e.g. 2017) with constant surface mixing ratio on a lower level, i.e. prescribing lower surface mixing ratio. We expect that the results of such a sensitivity simulation after ~50 years of simulation (e.g. 2065 for a start in 2017) in comparison with our SEN-C2-fCFC11_2050 (in 2050) will only change slightly quantitatively, but not qualitatively compared to the reference simulation. To our understanding the calculated ozone changes are primarily affected by the prescribed CFC-11 differences (e.g. between REF-C2 and SEN-C2-fCFC11_2050) rather than from the CFC-11 background value.

Although M18 hinted that the additional CFC-11 emissions might be released in eastern Asia (see also Rigby et al., 2019), we do not impose any specific regional features due to the long lifetime of CFC-11 (e.g. Rigby et al., 2013). In comparison with the reference simulation (REF-C2) the sensitivity simulation SEN-C2-fCFC11_2050 allows us to investigate the potential impact of previously unaccounted CFC-11 emissions. It enables a rough estimation of the additional possible ozone loss under constant CFC-11 surface mixing ratio in the coming years and how it may impact the timing of full recovery of the ozone layer. In addition, the second sensitivity simulation (SEN-C2-fCFC11_2020, see also Figure 1) is carried out, to allow for an estimate of ozone changes in case of full recognition and implementation of the Montreal Protocol again in the coming years. After an 18-year period of constant CFC-11 surface mixing ratio (in line with SEN-C2-fCFC11_2050 from 2002 to 2019), the CFC-11 surface mixing ratio is decreasing in the following years (in parallel with REF-C2) under the assumption that the recent additional CFC-11 emissions will drop down to zero starting in year 2020.

It is the aim of this study to show when and where ozone loss occurs due to additional influx of CFC-11 into the atmosphere while other ODSs decline as expected. Our scenario described in SEN-C2-fCFC11_2050 could be taken as "what should be avoided", somewhat in the tradition of the Newman et al. (2009) paper. Furthermore the SEN-C2-fCFC11_2020 provides an estimate of the impact of the temporary increase CFC-11 emissions in recent years on the ozone layer and its recovery. Our analysis presented here does not aim at specifically investigating the effects of the discovered CFC-11 emissions and numerous "directly" related scenarios. Here we want to assess the impact of enhanced CFC-11 surface mixing ratio as a

sensitivity study. The foundation for our numerical exercise is the REF-C2 simulation, which was performed with our CCM EMAC (see description in Section 2). The results of this reference simulation were one of our several contributions to the Dhomse et al. (2018) study and WMO (2018). The results of EMAC were checked against observation (for the past) in detail (e.g. Jöckel et al., 2016) and were also compared with results derived from other CCMs. No obvious weaknesses or significant deficiencies could be identified.

After a short description of the used CCM and the analyzed model simulations (REF-C2, SEN-C2-fCFC11_2050 and SEN-C2-fCFC11_2020) in the next section (Sec. 2), the CCM results are presented in Section 3. For the results, we focus on changes of total column ozone and partial stratospheric columns in specific geographical regions and seasons. To the best of our knowledge, a detailed ozone budget analysis for such sensitivity simulations is performed and presented for the first time. Finally, the discussion and conclusion are presented at the end of this paper.

## 2 Description of the model and simulations

For this study, the CCM EMAC (abbreviation stands for "European Centre for Medium-Range Weather Forecasts – Hamburg (ECHAM)/Modular Earth Submodel System (MESSy) Atmospheric Chemistry model") is used in the version 2.52 and is operated at a resolution of T42L90MA corresponding to a quadratic Gaussian grid of approx. 2.8 by 2.8 degrees in latitude and longitude with 90 levels up to 0.01 hPa. More details were presented by Jöckel et al. (2016).

The joint IGAC/SPARC Chemistry-Climate Model Initiative (CCMI) proposed several reference and sensitivity simulations for CCM studies. The aim was to support upcoming ozone and climate assessment reports. In this connection an internally consistent simulation from the past into the future between 1960 and 2100 has been suggested (Eyring et al., 2013). This transient reference simulation, i.e. REF-C2, as used in this study, is forced by trace gas projections and prescribed sea surface temperatures (SSTs) and sea ice concentrations (SICs). The projection component of REF-C2 uses greenhouse gas concentrations (i.e. $CO_2$, $CH_4$, and $N_2O$) that follow the Intergovernmental Panel on Climate Change (IPCC) Coupled Model Intercomparison Project Phase 5 (CMIP5) Representative Concentration Pathways 6.0 (RCP 6.0) scenario (van Vuuren et al., 2011). Monthly mean global SST and SIC data, which were simulated by the climate model HadGEM2 with an interactive ocean (Hadley Centre Global Environment Model version 2, data used for the RCP6.0 scenario; see Jones et al., 2011), were used as boundary conditions for the REF-C2 simulation.

For this study, in addition to the REF-C2 simulation (for more details see Jöckel et al., 2016; here it is the "RC2-base-04" reference simulation), two specific sensitivity simulations (SEN-C2-fCFC11_2050 and SEN-C2-fCFC11_2020) are designed to address the possible consequences of additional emissions of CFC-11, which affect the chlorine content of the stratosphere after some years. In all EMAC simulations mixing ratios of ODSs (CFCs: $CFCl_3$, $CF_2Cl_2$, $CH_3CCl_3$, $CCl_4$; HCFCs: $CH_3Cl$, $CH_3Br$; Halons: $CF_2ClBr$, $CF_3Br$) in the lowest model layer are adapted by Newtonian relaxation to observed or projected surface mixing ratios (Kerkweg et al., 2006). The applied tracer nudging procedure diagnoses the emission flux of CFC-11,

which is necessary to adjust to the prescribed surface mixing ratio. In the REF-C2 simulation the mean CFC-11 surface mixing ratio in the year 2002 is $258.3 \times 10^{-12}$ mol/mol (see Table 5A-3 in Daniel and Velders et al., 2011) and it is significantly reduced by more than 50% ($127.2 \times 10^{-12}$ mol/mol) in the year 2050 (i.e. the mixing ratio of the baseline (A1) scenario; WMO, 2011; the respective values are also presented in Figure 1). The 2050 value is projected under the assumption of full compliance with the Montreal Protocol. The cumulative CFC-11 emissions in the REF-C2 simulation (from 2002 to 2050) result in about 400 Gg. Apart from one point the sensitivity simulation (SEN-C2-fCFC11_2050) is identical to the reference simulation (REF-C2): in the sensitivity simulation the CFC-11 mean surface mixing ratio is kept constant at $258.3 \times 10^{-12}$ mol/mol after the year 2002, whereas they decline in REF-C2. The CFC-11 emissions required to achieve the constant surface mixing ratio value in our model after year 2002 in SEN-C2-fCFC11_2050 is about 90 Gg/year (e.g. for year 2003 it is 87 Gg/year; the emissions in our model simulation are slightly increasing with time, likely due to the small reduction of the CFC-11 lifetime; see for instance SPARC, 2013). The cumulative CFC-11 emissions (from 2002 to 2050) result in about 4500 Gg (i.e. roughly 4100 Gg more than in REF-C2). The emission values derived from observations given by Montzka et al. (2018) are about 65 Gg/year (mean) for 2002 to 2012 and 75 Gg/year from 2014 to 2016 (see also Rigby et al., 2019). The figures presented by Rigby et al. (2019) and Harris et al. (2019) with respect to the temporal evolution of CFC-11 emissions indicated a further increase after 2016. In the second sensitivity simulation SEN-C2-fCFC11_2020 after year 2020 full adherence of the Montreal Protocol is assumed. The starting point of this simulation is aligned with SEN-C2-fCFC11_2050 assuming the same constant CFC-11 surface mixing ratio between 2002 and 2019 (i.e. $258.3 \times 10^{-12}$ mol/mol), but with decreasing mixing ratios after 2020 until 2050. The cumulative CFC-11 emissions (from 2002 to 2050) result in about 2100 Gg (i.e. roughly 1700 Gg more than in REF-C2). A schematic of our three model simulations is presented in Figure 1.

In both sensitivity simulations we do not emphasize specific regions regarding outstanding changes of the CFC-11 surface mixing ratio, e.g. in eastern Asia (see discussion in the Introduction). The modified CFC-11 boundary condition in the SEN-C2-fCFC11_2050 and SEN-C2-fCFC11_2020 simulations should cause a change of the stratospheric chlorine loading after about 10-15 years (e.g. Engel et al., 2002).

## 3 Presentation of CCM results

### 3.1 Reactive chlorine

Based on the prescribed changes of the CFC-11 boundary conditions the stratospheric content of reactive chlorine compounds ($ClO_x$ = Cl + ClO + OClO + 2ClOOCl +2$Cl_2$ + HOCl) is expected to change. In Figure 2 the simulated change (i.e. SEN-C2-fCFC11_2050 and SEN-C2-fCFC11_2020 minus REF-C2, respectively) of reactive chlorine mixing ratios with time is shown for the lower stratospheric (LS, near 50 hPa) and the upper stratosphere (US, near 2 hPa). Because all model simulations are operated in a "free-running-mode" (i.e. not having the same meteorology), the year-to-year difference

(thin red and blue) curves as shown in Figure 2 (and also in other figures shown afterwards) are indicating the possible range of inter-annual fluctuation. Obviously it takes about 10-15 years of time (as expected) before the $ClO_x$ values of the REF-C2 and SEN-C2-fCFC11_2050 simulations clearly diverge from each other. At the end of the SEN-C2-fCFC11_2050 simulation (i.e. in 2050) the resulting absolute (mean) difference in EMAC arises to an approximately $6 \times 10^{-12}$ mol/mol

increase in the LS and about $50 \times 10^{-12}$ mol/mol in the US compared to the REF-C2 simulation. The vertical profile (structure) of the changes of chlorine mixing ratios qualitatively resembles reference $ClO_x$ profile in EMAC (i.e. changes are biggest where $ClO_x$ mixing ratios are biggest), which displays a distinct maximum at about 2-3 hPa (not shown). It turns out that in the US (above 30 km) in the 2040s (i.e. the time period from 2041 to 2050) the amount of chemically active chlorine species ($ClO_x$) in the SEN-C2-fCFC11_2050 simulation is on average about 17% larger than in the REF-C2 simulation; in

the LS (below 30 km) the respective amount of $ClO_x$ is enhanced by about 30%. As can be expected, the respective $ClO_x$ differences between the SEN-C2-fCFC11_2020 with respect to the REF-C2 simulation in the 2040s are clearly smaller, i.e. about $2 \times 10^{-12}$ mol/mol in the LS (about 14%) and about $25 \times 10^{-12}$ mol/mol in the US (about 9%).

### 3.2 Total and partial ozone columns

The impact of the enhanced atmospheric $ClO_x$ content due to constant CFC-11 surface mixing ratio after 2002 on total

column ozone (TCO) is shown in Figure 3. It illustrates the differences of the mean annual cycle for the last decade depending on the geographical latitude between the REF-C2 and the SEN-C2-fCFC11_2050 simulation. The largest changes of TCO are found in both polar regions: in particular in the Northern Hemisphere during winter (December, January, February) and in the Southern Hemisphere in late winter (August) to early spring (September, October). In the SEN-C2-fCFC11_2050 simulation the TCO values are clearly reduced by up to about 30 Dobson Units (DU) (in the order of 15% in

the Southern Hemisphere) in comparison to the REF-C2 simulation. During other times of the year and in other latitudinal regions the identified TCO changes are much smaller. They are mostly below ± 5 DU.

A closer look at the near global mean (60° S – 60° N) temporal behavior of TCO (Figure 4) indicates increasing ozone values in both simulations (upper part of the figure, for both simulations the solar cycle was removed, see figure caption). The results of the SEN-C2-fCFC11_2050 simulation are showing slightly smaller TCO values in comparison to REF-C2 and

a slightly flatter slope of the linear trend line (this regression accounts for possible auto-correlation at lag 1, see e.g. the method described in Tiao et al., 1990). The linear regression based on the results of the REF-C2 simulation (here the black line presented in the upper part of the figure) indicates an increase of 1.7 DU/decade for the TCO (annual near global mean), whereas the linear regression based on the results of the SEN-C2-fCFC11_2050 (red line in the upper part of the figure) shows a reduced increase of 1.3 DU/decade. This finding is supported by the TCO difference (lower part of the figure),

which indicates a small reduction of the TCO (in SEN-C2-fCFC11_2050) of about up to 2 DU (in the order of less than 1%) until 2050. The linear regression (again accounting for possible lag 1 auto-correlation) gives -0.5 DU/decade (± 0.25 DU/decade; given by two times the estimated standard error, which corresponds roughly to the 95% confidence interval and

will be used throughout this paper as a measure of uncertainty). This effect can be rated as negligible in comparison with the expected annual fluctuations in this region. Therefore in the following we focus on the analyses of the polar regions, in particular on the Antarctic region in September. One reason for choosing the month September for further analyses of Antarctic ozone chemistry is that we found here the most obvious ozone changes. Another reason is that this month is less

noisy compared to October (Solomon et al., 2016).

In the Southern Hemisphere polar region (70° S – 90° S) in September obvious ozone changes can be identified (Figure 5). The mean differences of the TCO between the 2000s and the end of the simulation amount to about 20 DU, indicating that the mean September ozone values in the SEN-C2-fCFC11_2050 simulation are about 10% lower than in the REF-C2 simulation. The trend estimate gives -4.1 DU/decade ($\pm$ 1.7 DU/decade). This trend estimate and the uncertainties have been

obtained by multiple linear regression, which accounts for possible lag-1 auto-correlation and uses the difference of the temperature anomalies (at 100 hPa over 70° S – 90° S) from the REF and the SEN simulation as second independent variable besides the linear trend. It is found that much of the inter-annual variability can be explained by including the difference of the polar temperature anomalies at 100 hPa in the regression model. This agrees with Langematz et al. (2016), who have used temperature anomalies to regress polar TCO. The temporal evolution of TCO differences and the size of the ozone

disturbance found in the Northern Hemisphere polar region in January have the same order of magnitude (not shown), but the signal is more noisy because of the stronger dynamic variability.

Now we are looking in more detail into stratospheric partial columns of ozone (PCO), for the upper stratosphere (US, above about 30 km, i.e. the 10 hPa pressure level) and the lower stratosphere (LS, between 100 hPa and 10 hPa) for the Antarctic region in September. Figure 6 shows the PCO differences for the US (top part) and LS (bottom part) between the SEN-C2-

fCFC11_2050 and the REF-C2 simulation. Both are showing the expected negative trend, indicating lower values at the end of the SEN-C2-fCFC11_2050 simulation in the late 2040s. Again the trends are obtained by the same regression model as for the TCO (see previous paragraph) but for the US the temperatures at 10 hPa have been used. The mean PCO changes for the US with about 2 DU are much smaller than those calculated for the LS (about 20 DU). The temporal evolution of the PCO differences in the LS show similar results as found for the TCO differences (Figure 5): in the SEN-C2-fCFC11_2050

simulation the TCO is reduced by about 20 DU until the year 2050. The strongest signature of ozone change found in the polar LS points to the importance of heterogeneous chemical processes. Viewing the vertical profile of the differences of net-ozone production rates of $ClO_x$ between the REF-C2 and the SEN-C2-fCFC11_2050 simulation in the 2040s clearly indicates an absolute minimum (i.e. less net-ozone production in SEN-C2-fCFC11_2050) at around 50 hPa and another relative minimum at about 1.5 hPa in September (Figure 8, right part; see explanation in Section 3.3). Complementary,

looking at the PCO changes in middle and lower latitude regions (60° S – 60° N, not shown) the partial column differences clearly indicate that the small ozone differences detected in the TCO (lower part of Figure 4) are affected by ozone reductions of similar magnitudes in the US and the LS, displaying only small contributions to the TCO.

In Figure 7 we are confronting the calculated differences between the simulations SEN-C2-fCFC11_2050 and REF-C2 as presented in Figures 4 to 6 with the corresponding results derived from SEN-C2-fCFC11_2020 and REF-C2. In all four parts of Figure 7 it is obvious that an immediate reimplementation of the Montreal Protocol eventually leads back to the direction of the expected ozone conditions around the end of the 2040s, as they are calculated in our REF-C2 simulation. In the REF-C2 simulation the model-diagnosed CFC-11 emissions are nearly zero after about 2030, whereas in the SEN-C2-fCFC11_2020 simulations they are steadily decreasing from higher values in 2020 down to zero around year 2050. This is caused by the prescribed CFC-11 surface mixing ratio as indicated in Figure 1.

## 3.3 Stratospheric ozone budget

In the following, a detailed analysis of individual ozone production and loss processes is carried out. This ozone budget analysis is used to investigate the role of separate chemical cycles and reactions, which are responsible for ozone production and loss in the stratosphere. For this analysis the MESSy tool StratO3Bud (cf. Meul et al., 2014, based on Jöckel et al., 2006) is employed. The respective reactions responsible for stratospheric ozone production (attributed to photolysis hυ, $HO_2$ and $CH_3O_2$) and loss (attributed to $O_x$, $NO_x$, $HO_x$, $ClO_x$ and $BrO_x$) are described by Meul et al. (2014, see their Table 2). In Figure 8 the results of this budget analysis are shown as changes of the ozone production rate between the SEN-C2-fCFC11_2050 and the REF-C2 simulation $\left(\Delta P_{SEN-REF}^{prc}\right)$ with respect to the total production rate in the REF-C2 simulation. The explicit formula for calculating the changes of the ozone production rate at a certain level is given as

$$\Delta P_{SEN-REF}^{prc}(lev) = \frac{\sum_{lat \in R} P_{SEN}^{prc}(lat, lev) - \sum_{lat \in R} P_{REF}^{prc}(lat, lev)}{\sum_{lev} \sum_{lat \in R} P_{REF}^{tot}(lat, lev)}$$

Here, P denotes the temporal mean and zonally summed ozone production rate (molecules/s). The subscript denotes the respective simulations and the "prc" superscript denotes, which process (hυ, $HO_2$, $CH_3O_2$, $O_x$, $NO_x$, $HO_x$, $ClO_x$, and $BrO_x$) is analyzed. Loss cycles are regarded as negative production rates. Further, the superscript "tot" denotes the sum of all positive production rates (namely of hυ, $HO_2$, $CH_3O_2$) and the summation goes over all latitudes, which lie in the respective latitudinal band R. Here we show profiles of $\Delta P_{SEN-REF}$ for the annual global mean and the Southern Hemisphere polar region (70°S – 90°S) during September 2041–2050. In the Antarctic spring season (Figure 8, right part) obviously the enhanced content of reactive chlorine in the SEN-C2-fCFC11_2050 simulation is responsible for the intensified ozone loss in the LS (around 50 hPa) and US (around 1.5 hPa). In the LS also ozone loss through the $BrO_x$ cycle is enhanced, probably related to the enhanced chlorine loading as ozone loss due to the reaction of BrO with ClO is attributed to the $BrO_x$ cycle (cf. Meul et al., 2014). On the other hand, as a consequence of more available chlorine, other loss cycles or production processes show a tendency to compensate the enhanced ozone destruction by chlorine. For instance the catalytic $NO_x$-cycle shows some balancing in the altitude region between about 19 km (50 hPa) and the stratopause (about 0.7 hPa). This means that in the 2040s the ozone depletion by $NO_x$ is clearly reduced (i.e. a relative ozone production) in the SEN-C2-fCFC11_2050

compared to the REF-C2 simulation above about 40 hPa. It is, however, not straightforward to further disentangle the underlying processes of the most relevant chemical cycles, because the underlying kinetic system is highly non-linear. The system in the SEN-C2-fCFC11_2050 simulation is heading towards a different chemical equilibrium, because the distribution of educts and the temperature change. The results with respect to global annual means of the ozone production and loss in the 2040s shown in Figure 8 (left part) indicate that below about 50 hPa no obvious changes are detected. Above 20 km the ozone loss is strongly affected by reactive chlorine and again some compensation effects in the US due to other competing ozone loss cycles are clearly identified. The positive values with respect to the photolysis rates indicate a slight downward shift of the ozone layer (ozone maximum) to lower altitudes. Probably, this is due to enhanced ozone loss through chlorine at higher altitudes, which allows more UV radiation to reach lower altitudes, where this additional radiation in turn causes higher photolysis rates.

The analogous ozone budget analysis is carried out for changes of the ozone production rate between the SEN-C2-fCFC11_2020 and the REF-C2 simulation. As expected, the vertical dependence of the ozone production rates in the 2040s looks similar as in the analysis of the SEN-C2-fCFC11_2050, namely (not shown): (1) the global annual means of the ozone production and loss below about 20 km (50 hPa) do not indicate obvious changes. Higher up ozone loss is strongly affected by reactive chlorine and again some compensation effects in the US due to other competing ozone loss cycles are clearly identified, but all ozone production rate changes are about half as strong as found between the SEN-C2-fCFC11_2050 and the REF-C2 simulation (see Figure 8 left part). (2) In the Antarctic region during September in the 2040s the intensified ozone loss in the LS (around 50 hPa) through the $ClO_x$ and $BrO_x$ cycles are again obvious, but the ozone destruction rates are only one third of the magnitudes, which are found between the SEN-C2-fCFC11_2050 and the REF-C2 simulation (see Figure 8 right part).

Finally, to check the possible impact of temperature changes due to enhanced CFC-11 concentrations on ozone chemistry we analyze the overall temperature trends in the US (near 1 hPa) and LS (near 50 hPa) and also the differences between REF-C2 and SEN-C2-fCFC11_2050. The global annual mean long-term temperature behavior in the REF-C2 simulation is indicating a cooling of about 1 K in the LS and of about 3 K in the US from the early 2000s until to the year 2050. The temperature difference between REF-C2 and SEN-C2-fCFC11_2050 amount to an additional cooling of about 0.3 K in the US in the SEN-C2-fCFC11_2050 simulation, whereas no obvious change in the long-term behavior can be identified in the LS (not shown). It is difficult to separate the individual contributions of the additional cooling in a coupled CCM simulation, i.e. radiative cooling by enhanced CFC-11 concentrations and by less ozone in the stratosphere caused by the enhanced chlorine loading without additional diagnostics. We assume that both processes will contribute to the calculated additional cooling in the SEN-C2-fCFC11_2050 simulation.

Taking a closer look to the Southern Hemisphere polar region in spring, the REF-C2 simulation is indicating a clear cooling trend of about 4 K in the US (near 1 hPa) until 2050, whereas no obvious trend can be identified in the LS (not shown). With respect to temperature differences of SEN-C2-fCFC11_2050 and REF-C2, the US does not show a clear change, whereas in

the LS the SEN-C2-fCFC11 simulation suggests some additional cooling from the early 2000s until 2050 by about 2 K. However, this difference is superposed by large inter-annual fluctuations.

With this in mind we can try to evaluate the calculated ozone differences in the sensitivity simulations in comparison to REF-C2. In the global mean US, on the one hand enhanced chlorine mixing ratios are leading to enhanced ozone depletion by the catalytic ozone destruction cycle; on the other hand the extra cooling is known to create a reduction of the ozone depletion rates by gas-phase chemistry (e.g. Haigh and Pyle, 1982). It turned out that the net effect here is slightly negative, i.e. indicate that ozone differences between REF-C2 and the sensitivity simulations in the US are dominated by the enhanced chlorine content. In the global mean LS, where no clear cooling is simulated, the smaller ozone values are therefore mainly caused by the enhanced chlorine content.

Looking closer at the South polar region in spring, it is obvious that in the US again the enhanced chlorine content is mostly responsible for the slightly reduced PCO in the SEN-C2-fCFC11_2050 simulation (Figure 6, upper part). In the LS, where heterogeneous chemical processes are the most important drivers of ozone changes, the enhanced chlorine mixing ratios intensify the ozone destruction. This leads to significantly reduced PCO over the time (Figure 6, lower part), which eventually leads to the indicated (slight) extra cooling of the polar lower stratosphere in spring. A first analysis of Polar Stratospheric Cloud (PSC) statistics for the REF-C2 and SEN-C2-fCFC11_2050 simulation displayed, however, no considerable trend in the PSC surface area (not shown). Therefore we cannot identify any hint for enhanced chlorine activation.

## 4 Discussions and Conclusion

After the detection of an unexpected and persistent increase in global emissions of CFC-11 (Montzka et al., 2018; see also update in Harris et al., 2019) it is still unclear (i) how much these additional emissions have already affected stratospheric ozone, (ii) how the CFC-11 emissions could further develop in the next years, and (iii) how large the potential for a disturbance of the temporal evolution of the ozone layer is. The discussions during the International Symposium in Vienna (March 2019) on the unexpected increase of the CFC-11 emissions came to the conclusion that a major problem is to create a realistic assessment of future CFC-11 levels (Harris et al., 2019). There are many factors, which have significant uncertainties, for instance the role of bank emissions or a possible co-production of CFC-12 ($CF_2Cl_2$) with CFC-11. There was a general acceptance that higher CFC-11 emissions are creating enhanced ozone depletion, but so far the corresponding magnitudes of ozone disturbances are uncertain. This study aims to estimate possible ozone changes due to enhanced CFC-11 values in recent and coming years, whereas the amount of other ODS are declining as expected. For that reason a simplified study based on CCM simulations is conducted first, to estimate roughly the implications of a constant mean CFC-11 surface mixing ratio for ozone depletion instead of reducing CFC-11 on longer time scales, and second, how strong the maximum ozone effect is due to the additional CFC-11 emissions in recent years. To keep things as simple as possible we do

not consider regional differences with respect to CFC-11 emissions in our sensitivity simulations. From our point of view considering regional differences would not have relevant effects on the presented results because of the long lifetime of CFC-11 (e.g. Rigby et al., 2013; Engel and Rigby et al., 2018), which leads to global mixing (Hoffmann et al., 2014). Our simplified approach is assuming an extreme boundary condition (i.e. constant CFC-11 surface mixing ratios until year 2050

in SEN-C2-fCFC11_2050 and until year 2019 in SEN-C2-fCFC11_2020), which is justified since currently a more realistic approach with respect to future CFC-11 levels is not available (Harris et al., 2019). Therefore, the presented results should not at all be taken as a robust prediction of future conditions.

The presented results indicate that mostly the ozone layer over the Arctic and Antarctic in late winter and spring is in particular affected by the prescribed CFC-11 surface mixing ratio change. In our case, at the end of the SEN-C2-

fCFC11_2050 simulation the impact on TCO culminates in a maximum ozone decrease of up to 30 DU in both polar regions (Figure 3). The calculated ozone changes at mid-latitude and tropical regions are surprisingly small (less than $\pm 5$ DU) and therefore are mostly not significant in the sense that the range of variability is in the same order of magnitude. An estimate of possible ozone changes in the late 2040s based on the perturbation of "Equivalent Effective Stratospheric Chlorine" (EESC) may lead to similar results, but appropriate explanations are lacking. Therefore, for the first time we perform a detailed

ozone budget analysis of such sensitivity simulations showing interesting results with regard to compensation and buffering effects associated with different production and loss cycles. It turns out from our results that the strengthened ozone depletion by enhanced chlorine is partly compensated by other ozone depleting catalytic cycles (e.g. $NO_x$) and other molecules (e.g. $HO_2$). For the global mean picture, there is no big TCO difference visible, as the effects of ozone production and loss processes are nearly cancelling. In the polar regions however, although there are also compensating effects, the

signal is noteworthy in spring (e.g. about 20 DU for the Antarctic region). We identify where (altitude) and at which time the ozone amount is decreased in the SEN-C2-fCFC11_2050 simulation compared to REF-C2. The ozone response to CFC-11 changes looks quasi linear, but the processes in the background are obviously non-linear.

Finally, based on the results of our SEN-C2-fCFC11_2050 simulation we try to approximately estimate the possible shift of the closure date of the ozone hole over Antarctica under the implied conditions of this sensitivity. For this we are looking at

the temporal evolution of the total stratospheric $ClO_x$ loading in the REF-C2 (started in the middle of the 20[th] century) and the SEN-C2-fCFC11_2050 simulation. In the REF-C2 simulation $ClO_x$ values are strongly increasing from 1960 onwards and are highest at the end of the 1990s. Starting in the 2000s the $ClO_x$ concentration is decreasing. The 1980 value of the REF-C2 simulation can be regarded as a reference for chlorine conditions before the ozone hole appeared. This "pre-ozone hole" chlorine content is reached again around the year 2050 in our REF-C2 simulation, which is some years earlier than the

multi-model mean based on all REF-C2 simulations as calculated by Dhomse et al. (2018). By extrapolating the linear regression line of the $ClO_x$ content (for 2002 to 2050) of the SEN-C2-fCFC11_2050 simulation (not shown) into the future, we estimate, that a pre-ozone hole, i.e. "1980", $ClO_x$ loading is likely to be reached before 2070. Therefore we can roughly determine a maximum delay of the closure of the ozone hole of somewhat less than about 20 years when keeping CFC-11

surface mixing ratio at a 2002 level vs. a decline of CFC-11 surface mixing ratio as it is reached through adherence of the Montreal Protocol. Considering that Dhomse et al. (2018) determined the closure date for the ozone hole by the year 2060 and that the one sigma standard deviation is in the range of about ±5 years (see also Figure 4.22 by Langematz and Tully et al., 2018), this indicates that the calculated effects of constant CFC-11 surface mixing ratio could have a non-negligible

effect on the closure date of the ozone hole. This finding is in line with other model results mentioned by Harris et al. (2019) that the closure of the ozone hole and ozone recovery as a whole will be delayed depending on the CFC-11 emission levels. A first estimate presented in WMO (2018) showed that if total CFC-11 emissions were to continue at levels experienced from 2002–2016 (67 Gg/year), the return of mid-latitude and polar EESC to the 1980 value would be delayed by about 7 years and 20 years, respectively. For the Arctic region enhanced stratospheric chlorine content means that there is the

possibility of stronger ozone depletion under specific dynamic conditions (i.e. a stable and cold polar vortex until March) for a slightly longer time period (e.g. Dameris and Godin-Beekmann et al., 2014).

The presented results do not show dramatic consequences for the global mean ozone layer due to enhanced CFC-11 surface mixing ratio for the next years, but indicate relevant changes in the polar regions in winter and spring. In the light of our results showing chemical feedback processes, which are diluting the effects due to enhanced CFC-11 levels in parts, the

compliance of the guidelines of the Montreal Protocol is absolutely necessary. Without a further strong regulation of the CFC-11 and other ODS emissions (e.g. Laube et al., 2014; Hossani et al., 2017), this could affect significantly the recovery of the ozone layer including the timing of the closure of the ozone hole – and this should be avoided!

*Code and data availability.* The Modular Earth Submodel System (MESSy) is continuously developed and applied by a consortium of institutions. The usage of MESSy and access to the source code is licensed to all affiliates of institutions, which are members of the MESSy Consortium. Institutions can become a member of the MESSy Consortium by signing the MESSy Memorandum of Understanding. More information can be found on the MESSy Consortium Web-site (http://www.messy-interface.org).

*Author contributions.* Both sensitivity simulations were set-up and carried out by P.J. with support of M.D.; M.D. structured and composed the manuscript. The author team analyzed jointly the model data and compiled the results and all three authors contributed to the manuscript.

*Competing interests.* The authors declare that they have no conflict of interest.

*Acknowledgements.* We would like to thank Heidi Huntrieser for her internal review and her useful suggestions to improve

the manuscript. Thank you also to two anonymous referees for their helpful comments on the paper. The work described in this paper has received some funding from the Initiative and Networking Fund of the Helmholtz Association through the

project "Advanced Earth System Modelling Capacity (ESM)". In addition, the authors acknowledge financial support from the DLR internal project KliSAW (Klimarelevanz von atmosphärischen Spurengasen, Aerosolen und Wolken), and the Research Unit SHARP by the Deutsche Forschungsgemeinschaft (DFG). The model simulations have been performed at the German Climate Computing Centre (DKRZ) support by the Bundesministerium für Bildung und Forschung (BMBF). We

used the NCAR Command Language (NCL, 2018) for data analysis and to create some of the figures of this study. NCL is developed by UCAR/NCAR/CISL/TDD and available on-line: http://dx.doi.org/10.5065/D6WD3XH5. CDO (Climate Data Operators; Schulzweida, 2019) was employed for processing the data. We furthermore thank all contributors of the project ESCiMo (Earth System Chemistry integrated Modelling), which provide the model configuration and initial conditions. We acknowledge the World Climate Research Programme's Working Group on Coupled Modelling, which is responsible for

CMIP, and we thank the HadGEM climate modelling group for producing and making available their model output. For CMIP the US Department of Energy's Program for Climate Model Diagnosis and Intercomparison provides coordinating support and led development of software infrastructure in partnership with the Global Organization for Earth System Science Portals.

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

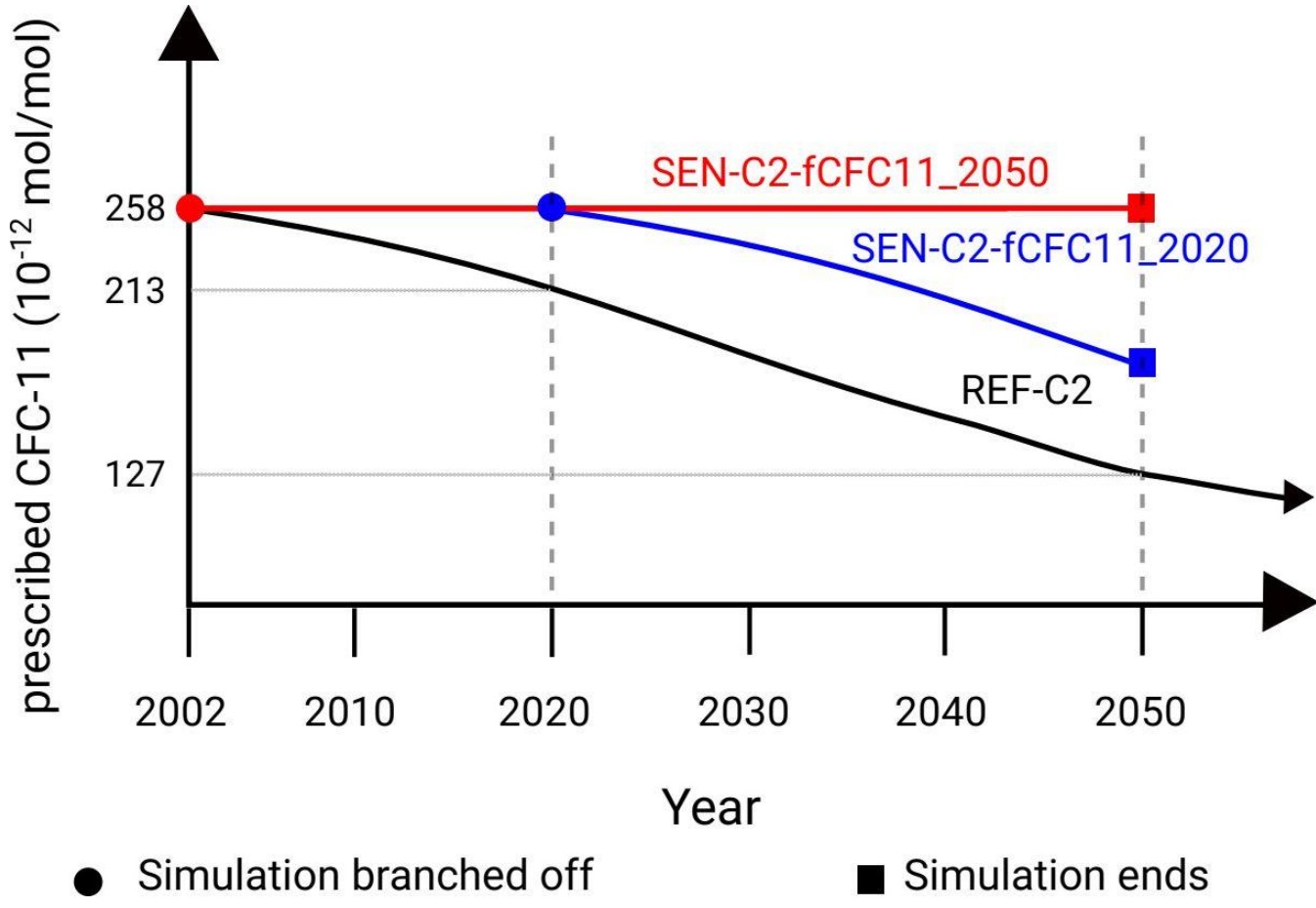

**Figure 1: Schematic of the performed EMAC model simulations: A reference simulation (REF-C2) and two sensitivity simulations (SEN-C2-fCFC11_2020 and SEN-C2-fCFC11_2050) enabling an assessment of enhanced CFC-11 surface mixing ratios on the ozone layer. The prescribed CFC-11 surface mixing ratios are given on the non-linear vertical axis. The prescribed CFC-11 surface mixing ratios are based on Table 5A-3 in Daniel and Velders et al. (2011).**

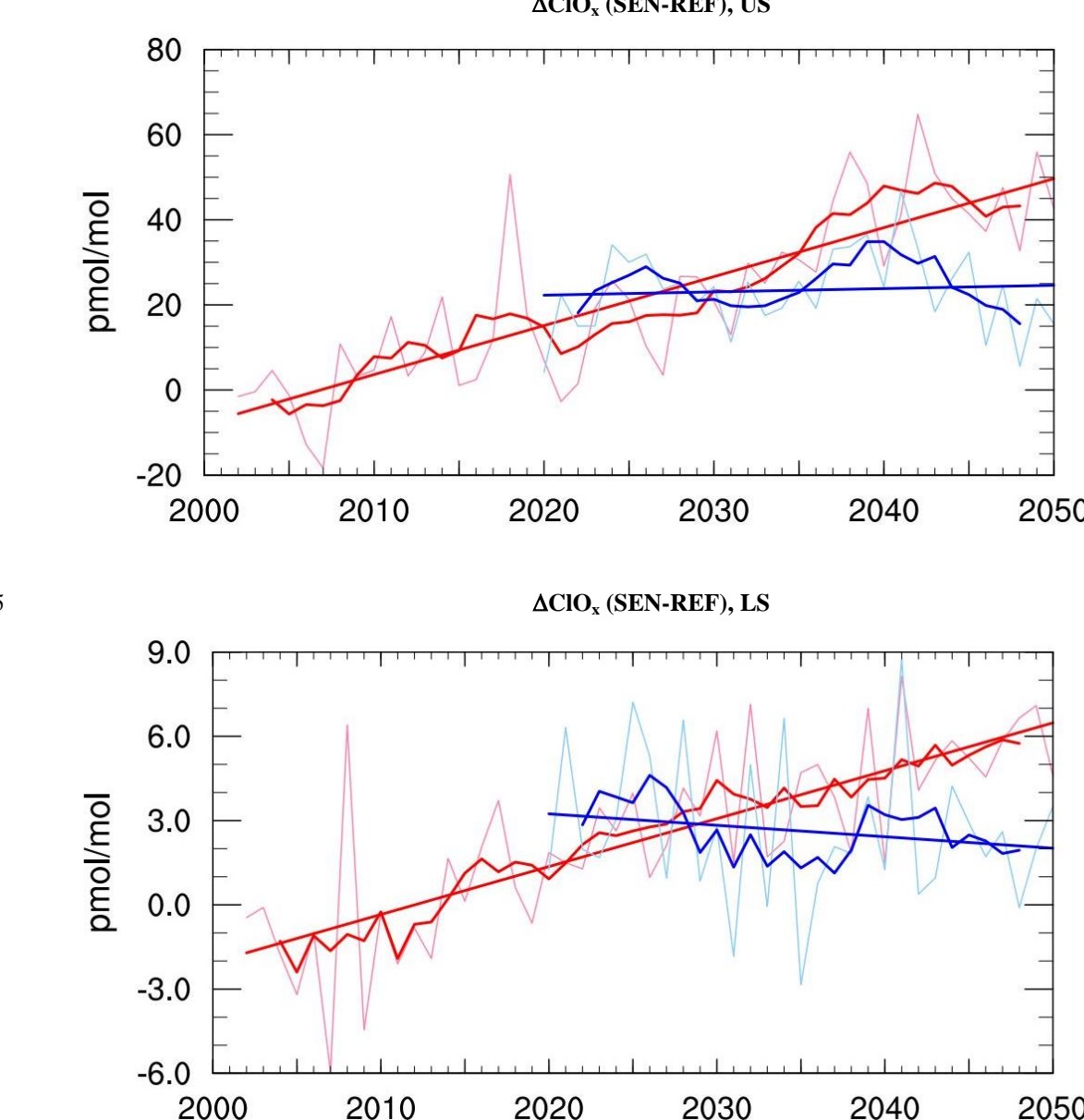

**Figure 2: Temporal evolution of the annual global mean ClO$_x$ mixing ratio differences (in mol/mol) at around 2 hPa (US, top) and 50 hPa (LS, bottom) between the SEN-C2-fCFC11_2050 and REF-C2 (in red) and between the SEN-C2-fCFC11_2020 and REF-C2 (in blue). The 11-year solar cycle (smoothed with a 1-2-1 filter) has been removed from both time series. The thicker curves in red and blue show the 5 year running means. The red and blue lines show the linear regression estimate of the unsmoothed time series.**

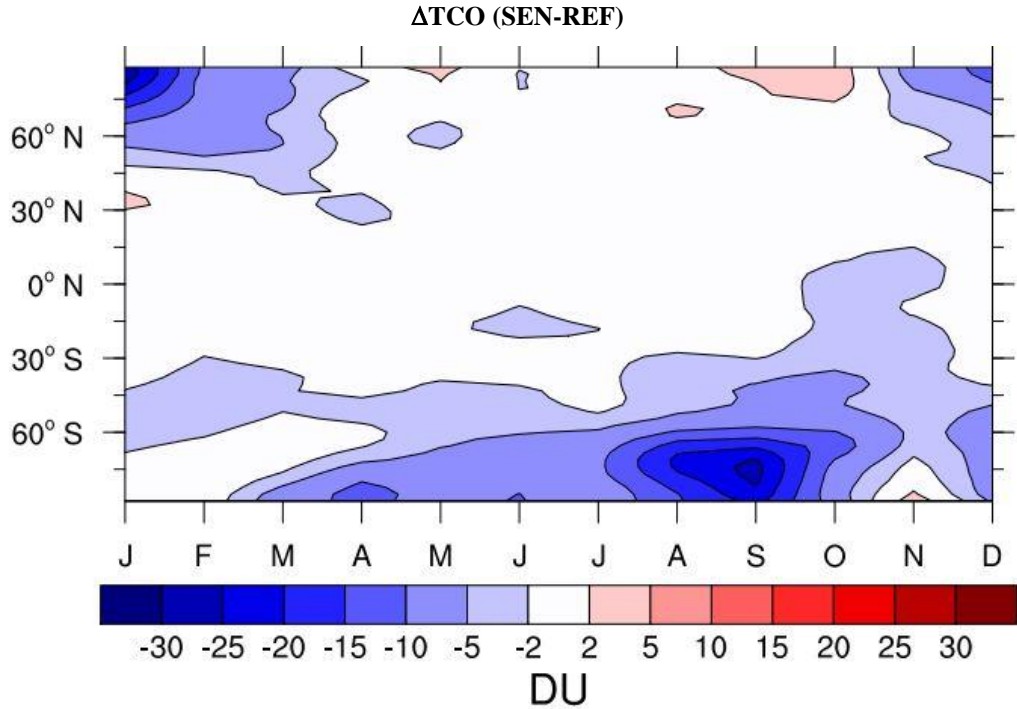

**Figure 3: Mean annual cycle of total column ozone (TCO) differences (in Dobson Units, DU) between SEN-C2-fCFC11_2050 and REF-C2 for the 2040s (i.e. SEN minus REF).**

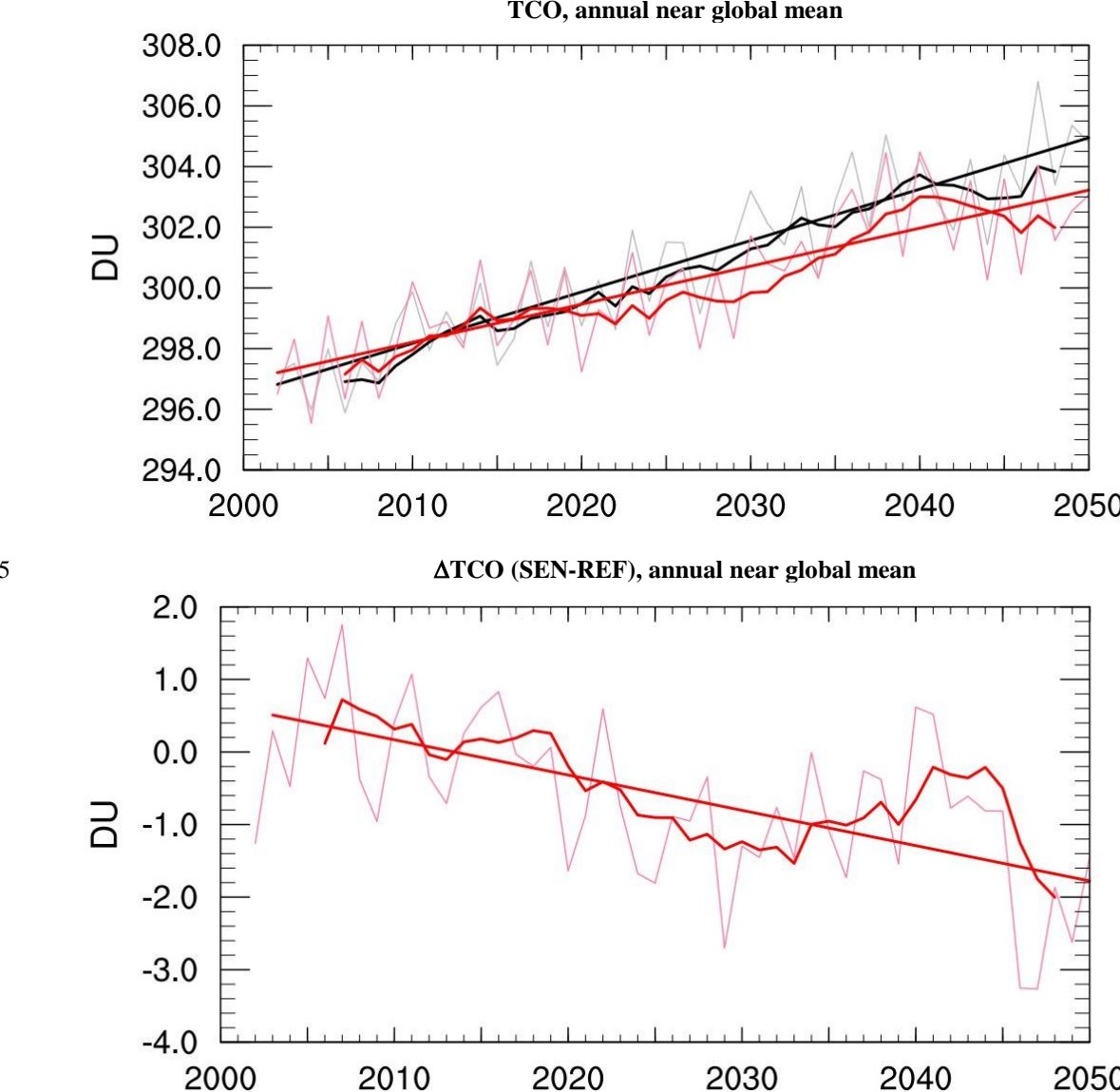

**Figure 4: Top: Temporal evolution of total column ozone (TCO; in DU) for the annual near global mean (60° S – 60° N) in REF-C2 (black curves) and in SEN-C2-fCFC11_2050 (red curves). Bottom: TCO differences (in DU) between SEN-C2-fCFC11_2050 and REF-C2 (i.e. SEN minus REF). For the absolute TCO time series (top) the 11-year solar cycle (smoothed with a 1-2-1 filter) has been removed. Thicker curves show the 5-year running means, respectively. The corresponding lines (top and bottom) show the respective linear regression estimates based on the unsmoothed data.**

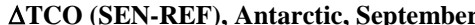

**ΔTCO (SEN-REF), Antarctic, September**

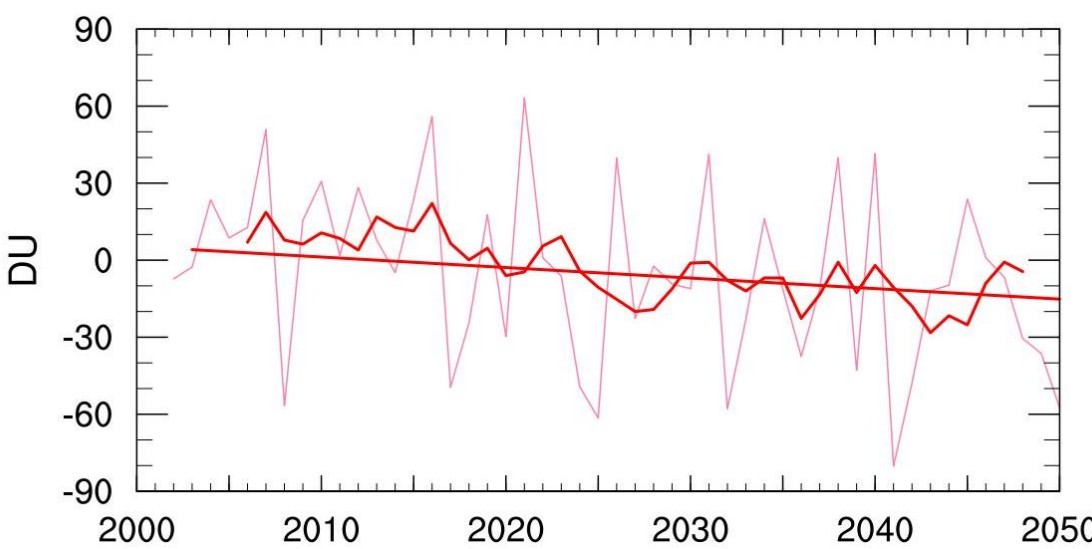

**Figure 5: Temporal evolution of TCO differences (in DU) between SEN-C2-fCFC11_2050 and REF-C2 (i.e. SEN minus REF) for the Antarctic region (70° S – 90° S) in September. The thicker curve shows the 5-year running mean. The corresponding line shows the trend estimate between for the unsmoothed time series using a multiple linear regression – including differences of temperature anomalies as dependent variable - which accounts for possible autocorrelation with lag 1 (see text for details).**

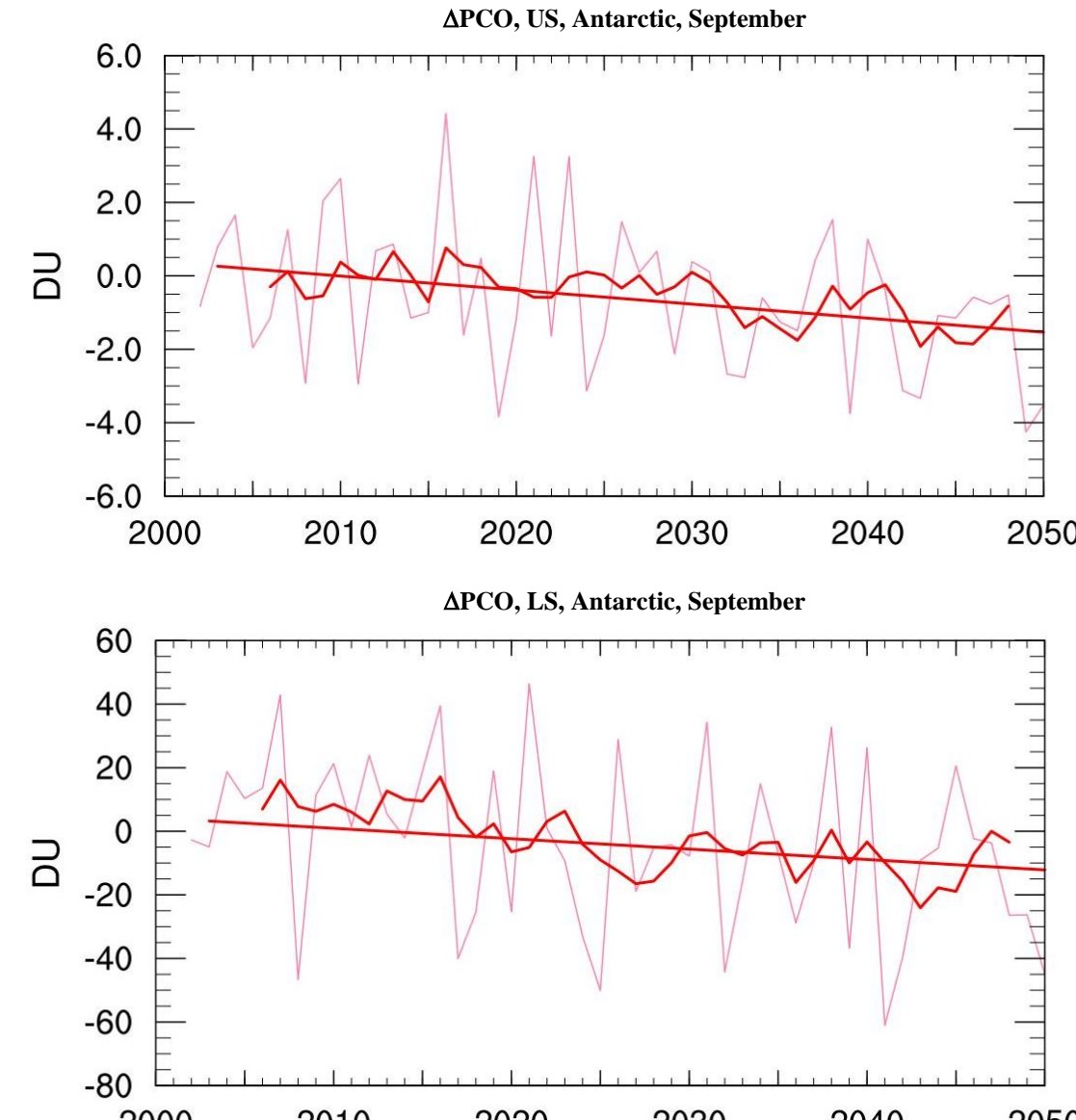

**Figure 6: Temporal evolution of Partial Column Ozone (PCO) differences (in DU) between SEN-C2-fCFC11_2050 and REF-C2 (i.e. SEN minus REF) for the Antarctic region (70° S – 90° S) in September. Top: PCO are shown for the US (above 30 km); bottom: PCO are shown for the LS (below 30 km). Red thicker curves show the 5-year running means, respectively. The corresponding lines show the trend estimates for the unsmoothed time series using a multiple linear regression – including differences of temperature anomalies as dependent variable – which accounts for possible autocorrelation with lag 1 (see text for details).**

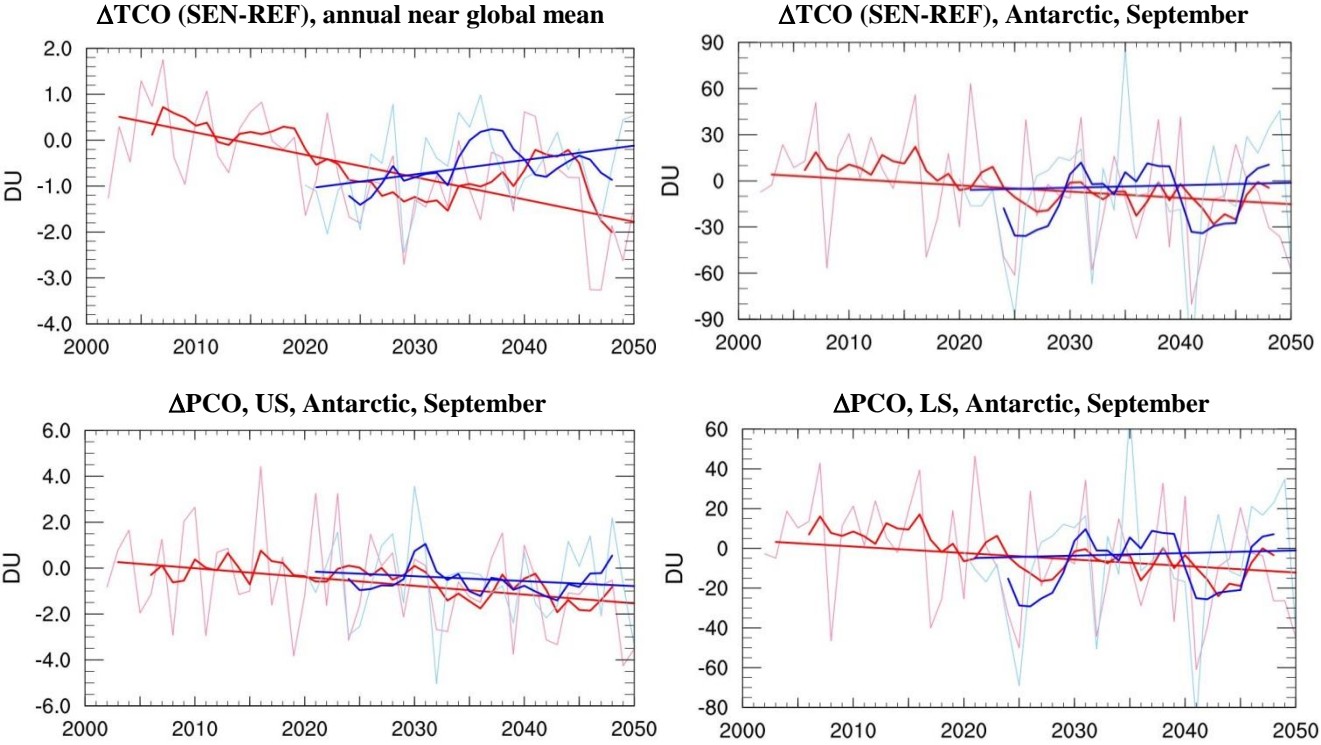

**Figure 7: Different temporal evolution of column ozone differences (in DU) between the individual sensitivity simulations and the reference simulation (in DU): SEN-C2-fCFC11_2050 minus REF-C2 values are indicated in red and SEN-C2-fCFC11_2020 minus REF-C2 in blue. Top left: TCO for the annual near global mean (60° S – 60° N); top right: TCO for the Antarctic (70° S – 90° S) in September; bottom left: PCO for the Antarctic (70° S – 90° S) in the US in September; bottom right: PCO for the Antarctic (70° S – 90° S) in the LS in September. The red and blue lines show the trend estimates for the unsmoothed time series using a multiple linear regression – including differences of temperature anomalies as dependent variable - which accounts for possible autocorrelation with lag 1 (see text for details).**

**Ozone production rate**

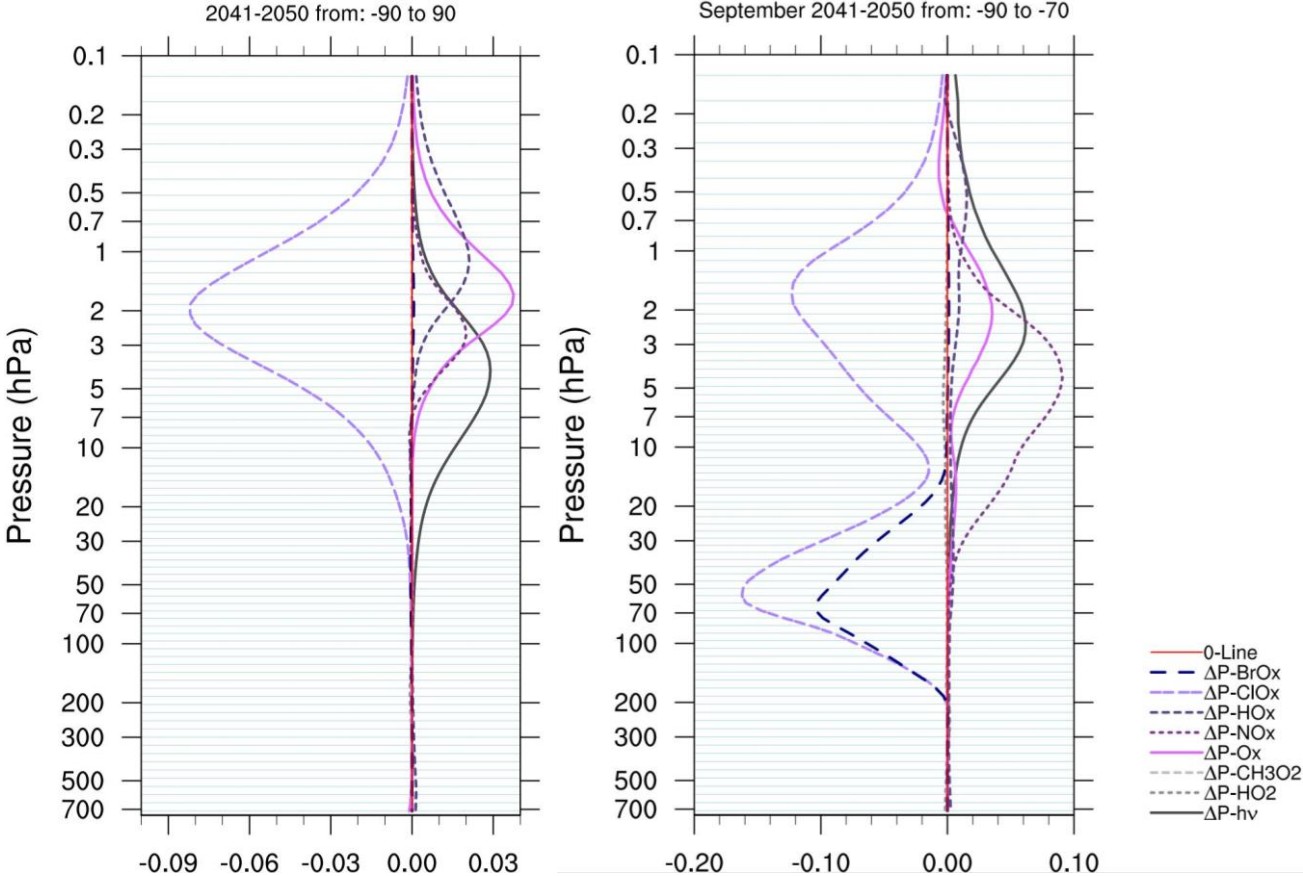

Figure 8: The relative change of ozone production rates (in %), which are normalized to the total column production (through photolysis hυ, $HO_2$ and $CH_3O_2$) in the REF-C2 simulation. For the individual ozone destruction cycles and molecules mean difference values are shown, which have been derived from the REF-C2 and the SEN-C2-fCFC11_2050 (i.e. SEN minus REF) simulation for the 2040s (from 2041 to 2050). Left: for the mean annual global mean profiles; right: for the South polar region (70° S – 90° S) in September. Negative values are indicating an intensified ozone loss or a decreased ozone production in the SEN-C2-fCFC11_2050 simulation, whereas higher values indicate more ozone production or less loss through a specific process. Thin horizontal lines indicate the nearest pressure levels to the model grid-boxes.