# Peer review of "Possible implications of enhanced CFC-11 concentrations on ozone"

_Atmospheric Chemistry and Physics, 2019_

## Referee Comment (RC1) · Anonymous Referee #1 · 20 Mar 2019

General comments:

The paper by Dameris and colleagues investigates the potential impact of enhanced CFC-11 concentrations on future ozone by means of a coupled chemistry-climate model. A recent publication by Montzka et al. showed that atmospheric CFC-11 concentrations have not declined as expected from the Montreal Protocol. Motivated by this finding Dameris et al. conducted a sensitivity simulation with the CCM EMAC for the first half of the 21st century assuming constant CFC-11 levels after the year 2002. This simulation has been compared to a reference in which atmospheric CFC-11 develops in compliance with the Montreal Protocol.

Although I like the idea of estimating the implications of increasing CFC-11 emissions on future ozone, I have major concerns related to the set-up of the performed sensi-

tivity simulation. I totally agree with the authors that the future evolution of CFC-11 is not know, and that therefore a simplified modelling approach may be justified, but assuming constant year 2002 concentrations was in my view the most unfortunate choice. We know from observations that atmospheric CFC-11 has further decreased since 2002, namely from about 258 ppt to about 230 ppt in 2017. That means that the sensitivity simulation assumes too high atmospheric CFC-11 for the time period 2002-2017. Thanks to its long lifetime this additional CFC-11 stays for a while in the atmosphere and makes a quantitative estimate of the recently discovered increased CFC-11 emissions for future ozone meaningless. I would have understood a sensitivity experiment that follows the observations until 2017 and assumes constant CFC-11 values afterwards.

In my opinion this study requires additional efforts before becoming acceptable for publication in ACP. Either the authors perform a new sensitivity study with a more meaningful set-up (not necessarily the one outlined above, if there are better ideas), or they have at least to provide an estimate of the overestimated increase in stratospheric chlorine loading due to fixed CFC-11 levels between 2002 and 2017 and the subsequent ozone loss in their current sensitivity run.

Specific comments:

- No matter which constant CFC-11 value the authors assume for their sensitivity, it would be interesting to estimate the emissions required to achieve or maintain these CFC-11 values. This would help to put the made assumptions into perspective, also with historic CFC-11 emissions, and to get an idea of how likely the chosen scenario is.

- In general I would appreciate to see some information on the statistical significance of the displayed differences between both model simulations.

- As this study is based on one CCM only, it would be interesting to see a short discussion about the sensitivity of ozone recovery and return dates in EMAC to stratospheric

chlorine compared to other CCMs, following the Dhomse et al. paper.

- Fig. 4, 5, and 6 show Antarctic ozone chemistry for September. Usually October is shown for Antarctic ozone. I assume the authors chose September because Fig. 2 shows the largest difference between both model simulations in September. Some explanation would be helpful.

- p4, l15-17: What is the percentage increase of ClOx in the LS?

- p6, l21-26: Do you attribute the discussed additional cooling in SEN-C2-fCFC11 to the additional CFC-11 or the changes in stratospheric ozone or both?

Technical corrections:

- p2, l9: Dohmse -> Dhomse

- p8, l15: Dohmse -> Dhomse

- Fig. 1, 3, 4, 5: I think one running mean, either 3 or 5 years, would be enough. Especially in Fig. 3 (top) the many different lines are rather confusing than helpful.

---

## Author Comment (AC1) · 9 Apr 2019

ACP-2019-239

**Implications of constant CFC-11 concentrations for the future ozone layer**

by Martin Dameris et al.

Reply to the referee #1 comments

Thank you to the referee for taking the time to review our manuscript. In the following we give our first answers regarding the points raised by the reviewer. The statements, comments and suggested corrections raised by the referees are printed in black and our comments are presented in blue. We will try to consider all of the raised points in the revised manuscript in an adequate manner.

Answers to referee #1

Thank you for your comments and suggestions to improve the manuscript.

General comments:

The paper by Dameris and colleagues investigates the potential impact of enhanced CFC-11 concentrations on future ozone by means of a coupled chemistry-climate model. A recent publication by Montzka et al. showed that atmospheric CFC-11 concentrations have not declined as expected from the Montreal Protocol. Motivated by this finding Dameris et al. conducted a sensitivity simulation with the CCM EMAC for the first half of the 21st century assuming constant CFC-11 levels after the year 2002. This simulation has been compared to a reference in which atmospheric CFC-11 develops in compliance with the Montreal Protocol. Although I like the idea of estimating the implications of increasing CFC-11 emissions on future ozone, I have major concerns related to the set-up of the performed sensitivity simulation. I totally agree with the authors that the future evolution of CFC-11 is not known and that therefore a simplified modelling approach may be justified, but assuming constant year 2002 concentrations was in my view the most unfortunate choice. We know from observations that atmospheric CFC-11 has further decreased since 2002, namely from about 258 ppt to about 230 ppt in 2017. That means that the sensitivity simulation assumes too high atmospheric CFC-11 for the time period 2002-2017. Thanks to its long lifetime this additional CFC-11 stays for a while in the atmosphere and makes a quantitative estimate of the recently discovered increased CFC-11 emissions for future ozone meaningless. I would have understood a sensitivity experiment that follows the observations until 2017 and assumes constant CFC-11 values afterwards.

We understand the concerns of the referee regarding the set-up of our sensitivity simulation, because the starting point of our simulation was the year 2002 instead of 2017. And yes, in these 15 years the mean surface CFC-11 mixing ratio decreased by 36,5 ppt (from 258,3 to 221,8 ppt) according to the prescribed scenario in the reference simulation. In the following we are trying to explain our approach hoping to convince the reviewer that her/his concerns are not justified.

The reason for starting the sensitivity simulation in the year 2002 was motivated by the statement of Montzka et al. that since the early 2000s there are obvious uncertainties with respect to the sources and sinks of CFC-11. Taking the CFC-11 mean surface mixing ratio of the year 2002 and a constant value until 2050 was primarily motivated by Figure 2a in Montzka et al. (2018) and also by Figure ES-2 in WMO (2019), because they showed stable emissions between 2002 and 2012, but also they clearly indicated an increase of the emissions after 2012 (until 2017). We are aware that stable emissions are not equal to stable concentrations. To work with a constant background value is of course a rough (crude) assessment of the future evolution of CFC-11. But since the future evolution of CFC-11

emissions (and also surface mixing ratio) is highly uncertain, such a simplified assumption of a constant surface mixing ratio is from our point of view certainly justified as an extreme scenario dating back to the point when expected emissions and observations started to diverge, and in particular because of the lack of more precise information about future evolution of CFC-11 emissions.

If we would follow the suggestion by ref #1, we are convinced that most of our results may only slightly change quantitatively, but not qualitatively. Our argument line is the following:

Our investigations are focusing on differences (!) between the REF and SEN simulations. To our understanding, starting the SEN simulation in the year 2017 (instead of 2002) and running it until 2065 (instead of 2050) would lead to similar results.

In Table 5A-3 (WMO, 2011) the mean surface CFC-11 mixing ratios (and used in our model) are given as follow:

- 2002: 258,3 ppt
- 2015: 227,6 ppt
- 2017: 221,8 ppt (not about 230 ppt as pointed out by ref #1)
- 2020: 213,0 ppt
- 2050: 127,2 ppt
- 2065:   94,4 ppt

In our presented work we have used the surface mixing ratio for the years 2002 and 2050, which means that we have reduced the CFC-11 surface value in this time period by about 50%, i.e. **131,1 ppt difference** between the REF and SEN simulation **in 2050**.

Following the suggestion by the referee this would mean: The alternative sensitivity simulation SEN would be started in 2017 (with a constant surface mixing ratio of 221,8 ppt, as calculated from Table 5A-3) and this simulation would be performed until the year 2065. This would result in a difference of the CFC-11 surface mixing ratio of **127,4 ppt in 2065** (SEN minus REF). It turned out that this CFC-11 difference value is similar to the one which was calculated in our work (difference of 131,1 ppt in 2050). Therefore the calculated and presented differences of the TOC and PCO changes (and also the results of the ozone budget changes) in our investigation should be very similar to the results of the alternatively suggested simulation by the referee. To our understanding the calculated ozone changes (SEN minus REF) are primarily affected by the prescribed CFC-11 differences (between REF and SEN) rather than from the CFC-11 background value, which is of course different (by 36,5 ppt at the surface) in 2002 and 2017.

From our point of view one could argue in our paper that our model set-up can be taken as an upper limit estimate (like an extreme scenario) regarding the CFC-11 background condition. We will prepare an appropriate formulation in the revised manuscript.

In my opinion this study requires additional efforts before becoming acceptable for publication in ACP. Either the authors perform a new sensitivity study with a more meaningful set-up (not necessarily the one outlined above, if there are better ideas), or they have at least to provide an estimate of the overestimated increase in stratospheric chlorine loading due to fixed CFC-11 levels between 2002 and 2017 and the subsequent ozone loss in their current sensitivity run.

In light of our statements from above, we refrain from performing a new sensitivity study (e.g. covering 2017-2065 with fixed concentrations after 2017). We hope that the presented arguments

regarding the similarity of the differences in the prescribed surface mixing ratios and related differences in stratospheric chlorine show that such an additional sensitivity is likely to influence the majority of our results only on a quantitative basis and not qualitatively. This is the case as we are focusing mainly on differences between the REF and the SEN which are mainly depending on the differences in the chlorine loading.

Specific comments:
- No matter which constant CFC-11 value the authors assume for their sensitivity, it would be interesting to estimate the emissions required to achieve or maintain these CFC-11 values. This would help to put the made assumptions into perspective, also with historic CFC-11 emissions, and to get an idea of how likely the chosen scenario is.

We can certainly estimate the exact emissions which are required to achieve or maintain our corresponding CFC-11 surface values. They will be presented and discussed in more detail in the revised manuscript.

A quick look on our model data output, the following value has been estimated:

The calculated CFC-11 emissions in the CCM EMAC (because of the prescribed surface mixing ratio), which are needed to keep the surface CFC-11 conditions stable on the value of 2002, turned out to be of the order of 87 Gg/a in the year 2003.

The emissions presented in Figure ES-2 (WMO, 2019; based on Figure 2a in Montzka et al., 2018) indicate mean CFC-11 emissions of 65 Gg/a, for the period from 2002 to 2012; for the years from 2014 to 2016 the CFC-11 emissions are around 75 Gg/a.

- In general I would appreciate to see some information on the statistical significance of the displayed differences between both model simulations.

Thank you for this suggestion. We will prepare and present some more information of the significance of our results in the revised manuscript! We did not focus on statistical significance as we know the general processes behind the additional ozone loss, i.e. we know that we put in additional chlorine which will change ozone abundances.

- As this study is based on one CCM only, it would be interesting to see a short discussion about the sensitivity of ozone recovery and return dates in EMAC to stratospheric chlorine compared to other CCMs, following the Dhomse et al. paper.

Will be considered in the revised manuscript!

- Fig. 4, 5, and 6 show Antarctic ozone chemistry for September. Usually October is shown for Antarctic ozone. I assume the authors chose September because Fig. 2 shows the largest difference between both model simulations in September. Some explanation would be helpful.

We will add additional explanations in the revised manuscript. Yes, we chose September because it shows the largest ozone changes. But also because September is less noisy compared to October (see e.g. Solomon et al., Science, 353, Issue 6296, 269-274, doi: 10.1126/science.aae0061, 2016).

- p4, l15-17: What is the percentage increase of ClOx in the LS?

Will be calculated and discussed in the revised manuscript.

- p6, l21-26: Do you attribute the discussed additional cooling in SEN-C2-fCFC11 to the additional CFC-11 or the changes in stratospheric ozone or both?

This will be explained and briefly discussed in the revised manuscript. It is difficult to determine the individual contributions of the additional cooling in a coupled CCM. We assume that both processes (i.e. radiative cooling by enhanced CFC-11 concentrations and by less ozone in the stratosphere caused by enhanced chlorine loading) will lead to the calculated additional cooling in the SEN simulation.

Technical corrections:
- p2, l9: Dohmse -> Dhomse
- p8, l15: Dohmse -> Dhomse

Of course! Sorry.

- Fig. 1, 3, 4, 5: I think one running mean, either 3 or 5 years, would be enough. Especially in Fig. 3 (top) the many different lines are rather confusing than helpful.

Thank you for the suggested correction regarding the figures. Will be changed!

---

## Referee Comment (RC2) · Anonymous Referee #2 · 19 May 2019

This paper details the effect of a very specific future scenario of CFC-11 on ozone recovery. It is a response to the recent measurements showing the CFC-11 values are not dropping as quickly as predicted by compliance to the Montreal Protocol and thus implying illegal emissions. It outlines this one simple scenario in sufficient detail and the paper is well written. However, it is severely lacking in answering the questions necessary to understand the effect of the newly discovered emissions (see below for details). Thus, I cannot recommend publication of this paper in anything like its present form. I suggest the authors rethink the scope of the problem and expand their study considerably.

General Comments

The recent paper of Montzka et al. 2018 makes a strong case that there are illegal

emissions of CFC-11 presently occurring and that these have been occurring since 2012 and even perhaps earlier. This brings up many questions for future ozone recovery including (but not limited to): Have the emissions to date affected future ozone recovery; How much more would continuation of the present emissions to various end dates affect ozone recovery; What if the emission increased, what would that do to future recovery; What if there are banks of CFC-11 (and perhaps CFC-12) associated with the illegal emission; etc. The present paper does not address these questions in any detail. Instead it only addresses one simple scenario: if the mixing ratio of CFC-11 stays constant through 2050, what is the effect on ozone. This assumes that the emission rate of CFC-11 stays at a constant level slightly higher than any inferred emission estimated in Montzka et al. and that this emission stays constant for the next 3 decades. Ignoring that this scenario is unlikely to occur given the international response to this issue, the real problem with the paper is that so little of the problem space is explored. I see limited value in modeling one (unlikely) scenario in detail and ignoring all other possibilities.

I can only assume this choice was made because it is easy to implement in their model and it only took one new run. Unfortunately, the only question answered is that if a larger emission than inferred in Montzka et al. is continued for three decades it will have a negative effect on ozone recovery. This will surprise no one and in fact it can be predicted by computing the perturbation of EESC in 2050 by changing the CFC-11 mixing ratio between the ref value to the new value. This is a "back of the envelope" calculation. I expect much more of the problem space explored in a paper addressing the effect of illegal emissions of CFC-11 on ozone recovery and with a chemistry-climate model to use. As stated above, I recommend that the author team rethink the issues the Montzka et al. paper uncovered and take a real stab at helping answering them. It is necessary to frame the problem in terms of various possibilities for the emissions (and bank changes) and then from there predict mixing ratio scenarios and finally model time series. Only then can the reader understand the scope of the problem and the possible effects on future ozone.

---

## Author Comment (AC2) · 29 May 2019

ACP-2019-239

**Implications of constant CFC-11 concentrations for the future ozone layer**

by Martin Dameris et al.

Reply to the referee #2 comments

In the following we give our first reply regarding the points raised by the reviewer. The statements and comments given by the referee are printed in black and our comments are presented in blue.

Answers to referee #2

We thank the referee for taking time to review this manuscript and for the comments. Before we will answer the points raised by the referee on a point to point basis, we want to add a general explanation to set our work into context.

Our analysis presented here is not aimed at specifically investigating the effect of the newly discovered emission and numerous possible "directly" related scenarios. Here we want to assess the impact of enhanced CFC-11 concentrations on the ozone layer as a sensitivity study. The recent paper by Montzka et al. (2018) triggered our interest in investigating this CFC-11 sensitivity and justifies (to some extent) such a sensitivity study as an extreme sensitivity. In particular we think that our sensitivity is new and interesting as

- CFC-11 concentrations are stable whereas other ODS decline as expected;

- we show when and where O3 loss due to additional CFC-11 occurs;

- we perform an ozone budget analysis showing compensation and buffering effects associated with different production and loss cycles (this has not been shown before);

- we show that even with these extreme CFC-11 concentrations it takes time to see the effect on O3 (due to strong interannual variability; this also implies that in addition to long-term monitoring of O3 the monitoring of ODS is crucial); and

- we can provide (most likely) an upper limit for possible scenarios; this could be seen as "what should be avoided/what could happen as a worst case (e.g. if a lot of the new production is actually stored in banks)" somewhat in the tradition of the Newman et al. (2009) paper.

We will try to make these points more clearly in the revised version of the manuscript.

Preface:

This paper details the effect of a very specific future scenario of CFC-11 on ozone recovery. It is a response to the recent measurements showing the CFC-11 values are not dropping as quickly as predicted by compliance to the Montreal Protocol and thus implying illegal emissions. It outlines this one simple scenario in sufficient detail and the paper is well written. However, it is severely lacking in answering the questions necessary to understand the effect of the newly discovered emissions (see below for details). Thus, I cannot recommend publication of this paper in anything like its present form. I suggest the authors rethink the scope of the problem and expand their study considerably.

The referee is right in saying that our model approach is simple. However, there is also a big knowledge gap regarding the future evolution of CFC-11 emissions and our main aim is not to distill specific scenarios. Nevertheless we believe that on the basis of our model study we can answer relevant questions which are related to the effects of the additional CFC-11 emissions (i.e. constant CFC-11 concentrations). From our point of view the model simulation (SEN) which has been

performed in addition to the reference simulation (REF) is certainly an extreme sensitivity study regarding CFC-11. Hopefully, it serves as an upper limit of possibilities regarding future CFC-11 emissions. It should not be considered as a specific future scenario that we deem likely. This study aims to assess the magnitude of possible ozone changes under constant CFC-11 concentrations. Since currently we do not have more detailed information about possible future CFC-11 emissions (either from illegal production or banks) such a "simple sensitivity study" is in our understanding justified. The goal of this paper is to answer questions which are related to the newly discovered emissions, but we do not claim that this is the actual path we are on. Moreover, this paper was not intended to answer questions which are focusing on the origin or sources of the additional CFC-11 emissions.

Furthermore, we do not aim at making a complete assessment. Our study rather aims to provide an insight into the reaction of the ozone layer with respect to unchanged (stable) CFC-11 concentrations in the coming years. (Further, once a new "most likely" emission scenario is agreed on we are willing to provide such a simulation.)

Last but not least, it is an interesting sensitivity study to keep CFC-11 constant, whereas other ODS are declining according to RCP scenarios (this is new and interesting in itself). To fully carve this out, we also investigate in detail ozone loss and production rates.

General comments:

The recent paper of Montzka et al. 2018 makes a strong case that there are illegal emissions of CFC-11 presently occurring and that these have been occurring since 2012 and even perhaps earlier. This brings up many questions for future ozone recovery including (but not limited to):

With our model study based on a comparison of two simulations using a chemistry-climate model (here EMAC) we would like to provide a qualitative and also a rough quantitative assessment of possible stratospheric ozone changes due to unchanged conditions regarding CFC-11 concentrations. It should help to assess the correct order of magnitude of ozone changes, which can be directly related to additional CFC-11 emissions.

Have the emissions to date affected future ozone recovery;

This question can be roughly answered on the basis of the available model data. Thank you for this question and we will discuss this point in more detail in a revised manuscript. Certainly the additional emissions until today will have an impact on the current and future ozone values; but these values are definitely much smaller than those derived from the calculated ozone changes (i.e. SEN minus REF).

How much more would continuation of the present emissions to various end dates affect ozone recovery;

This question can be similarly answered as the question before. Our suggestion is that we will discuss this point in more detail for instance in the final section of the paper. Here we could discuss more precisely the range of uncertainty, both regarding future evolution of CFC-11 emissions and also with respect to our model strategy.

What if the emission increased, what would that do to future recovery;

As said in the paper (and also mentioned above) answering such a question is difficult and in parts speculative. If the CFC-11 emissions will increase (or hopefully decrease) this will certainly shift the date of full recovery. To answer this question in more detail would at least require two more

sensitivity model simulations (with higher or lower implied emission rates than in our SEN simulation), but the possible rates would be also arbitrary. Our sensitivity simulation can be taken probably as an upper limit; since the CFC-11 amount stored in banks is also not known, our sensitivity simulation with constant CFC-11 concentrations is not completely unrealistic; so with our REF and SEN simulations we are analyzing an extreme scenario.

What if there are banks of CFC-11 (and perhaps CFC-12) associated with the illegal emission;

As stated before, answering such a specific question (which is of course very interesting but highly speculative) would require more model simulations and would definitely burst our investigation. Since there are surely several other "if"-questions, which are interesting and waiting for answers, it is not our intention to work on all related questions. Our intention is focusing on the ozone changes due to constant CFC-11 concentrations.

The present paper does not address these questions in any detail. Instead it only addresses one simple scenario: if the mixing ratio of CFC-11 stays constant through 2050, what is the effect on ozone. This assumes that the emission rate of CFC-11 stays at a constant level slightly higher than any inferred emission estimated in Montzka et al. and that this emission stays constant for the next 3 decades.

It is not our intention to answer such questions as already pointed out. Our simulation was created in a way that the implied emissions are at the upper end of possibilities and therefore can be taken as an upper boundary for the assessment of ozone values. Certainly many other scenarios are possible and in principle could be "calculated" with our model; the number of possible future CFC-11 scenarios is arbitrarily high. Therefore this would be an endless story because many model simulations have to be carried out. In addition it is obvious that results might depend also on the specific model which is used. Thus, these questions could ultimately only be answered with a multitude of scenarios AND a multitude of models; this is clearly beyond the scope of our work!

From our point of view our assessment based on only two model simulations provides reliable information regarding the reaction of the stratospheric ozone with respect to unchanged CFC-11 concentrations and changes in the ozone production and loss cycles.

Ignoring that this scenario is unlikely to occur given the international response to this issue, the real problem with the paper is that so little of the problem space is explored. I see limited value in modeling one (unlikely) scenario in detail and ignoring all other possibilities.

As mentioned before, we take our sensitivity study as an upper boundary of possibilities. In addition to the arguments given above, we have also to keep in mind the computational costs for such a more extensive chemistry-climate model exercise. Other model systems, e.g. 2d models or models with a simplified chemistry scheme, might be more appropriate for such investigations with a larger amount of sensitivity studies.

I can only assume this choice was made because it is easy to implement in their model and it only took one new run. Unfortunately, the only question answered is that if a larger emission than inferred in Montzka et al. is continued for three decades it will have a negative effect on ozone recovery.

Due to the lack of more detailed information regarding the CFC-11 emission and its future evolution we decided to carry out only this model simulation under unchanged CFC-11 concentrations, which is the most simplified assumption you can make. As said already, here we try to investigate an extreme, or (hopefully) worst case scenario; it is certainly an interesting question as all other CFCs are assumed

to decrease. Our main interest is related to the question: "What must be avoided?" (a little bit like the Newman et al. (2009) "World avoided" paper).

This will surprise no one and in fact it can be predicted by computing the perturbation of EESC in 2050 by changing the CFC-11 mixing ratio between the ref value to the new value. This is a "back of the envelope" calculation.

In Section 3.3 (Stratospheric ozone budget) we have prepared and presented a detailed analysis of the model results with respect to the ozone production and loss cycles. And there are some "surprises", for instance that the effects are smaller than expected (as it will be calculated back of the envelope by using the EESC) due to some compensating effects by other ozone production and loss cycles. Our budget analysis nicely shows compensating effects for the global annual mean and in the upper stratosphere in the SH polar region, whereas in the lower stratosphere in the SH polar region the compensation does not occur. We show, which processes are buffering the additional O3 destruction through higher CFC-11 concentrations. We show where (altitude) and at which time O3 is decreased compared to the REF; these results are not easily obtained "back of the envelope"! Further we show that the additional emissions need time to affect the O3. It would have taken years to register some deviation in the O3 columns due to the additional emissions! And we have analyzed the effect of the temperature change on the chemistry.

I expect much more of the problem space explored in a paper addressing the effect of illegal emissions of CFC-11 on ozone recovery and with a chemistry-climate model to use. As stated above, I recommend that the author team rethink the issues the Montzka et al. paper uncovered and take a real stab at helping answering them. It is necessary to frame the problem in terms of various possibilities for the emissions (and bank changes) and then from there predict mixing ratio scenarios and finally model time series. Only then can the reader understand the scope of the problem and the possible effects on future ozone.

The chosen configuration of this model study has been used because we do not have more detailed information about future emissions available. Further, our study was triggered by the Montzka et al. (2018) paper, but we do not claim that our sensitivity represents the actual pathway of future emissions/concentrations. Instead we state, that it is most likely an extreme sensitivity study. A solution for our paper would be that in the final section (Discussion and conclusion) we will bring up some of the uncertainties (i.e. the CFC-11 emissions and also with respect to our model strategy) and make a more general discussion at the end which allows the reader to put our model study into the right context.

We very much hope that the referee can follow our argumentation line.

---

## Author Response (AR1)

ACP-2019-239

Old title: **Implications of constant CFC-11 concentrations for the future ozone layer**

Revised title: **Possible implications of enhanced CFC-11 concentrations on ozone**

by Martin Dameris et al.

Reply to the referee #1 comments

We thank the referee for taking the time to review our paper. In our revised manuscript we have considered all remarks and statements, which were raised in this expert's report. The paper has undergone significant changes. A second sensitivity simulation has been performed to enhance the scientific statement of our investigation. The title of the paper has been changed accordingly.

Among other modifications, we now provide a much clearer explanation about what the objectives of this study are. Apart from this, it is our intention to make it clearer for what purpose this study should not be taken, i.e. that it is not a robust projection of future ozone changes under different conditions regarding CFC-11. In the meantime, our strategy regarding our CCM sensitivity simulations received some support: The statements given by Harris et al. (2019; SPARC newsletter, summary of the CFC-11 workshop in Vienna, March 2019) are very clear in the sense that more concrete and more realistic projections of future CFC-11 levels are currently not possible. Therefore our approach is justified, allowing a simplified ("worst case") assessment of ozone changes due to enhanced CFC-11. Our second sensitivity study has been performed to provide a clearer picture with respect to the question on how much the recent CFC-11 changes so far have already affected ozone and how important the consequences are.

In the following we give our answers regarding all individual points raised by the reviewer. The statements, comments and suggested corrections raised by the referee are printed in black and our comments are presented in blue. If possible, we are pointing to the corresponding revised text block or paragraph. The revised manuscript including all changes (highlighted) and a cleared version (without highlighted changes) are provided.

Answers to referee #1 and our explanations

General comments by reviewer #1:
The paper by Dameris and colleagues investigates the potential impact of enhanced CFC-11 concentrations on future ozone by means of a coupled chemistry-climate model. A recent publication by Montzka et al. showed that atmospheric CFC-11 concentrations have not declined as expected from the Montreal Protocol. Motivated by this finding Dameris et al. conducted a sensitivity simulation with the CCM EMAC for the first half of the 21st century assuming constant CFC-11 levels after the year 2002. This simulation has been compared to a reference in which atmospheric CFC-11 develops in compliance with the Montreal Protocol. Although I like the idea of estimating the implications of increasing CFC-11 emissions on future ozone, I have major concerns related to the set-up of the performed sensitivity simulation. I totally agree with the authors that the future evolution of CFC-11 is not known and that therefore a simplified modelling approach may be justified, but assuming constant year 2002 concentrations was in my view the most unfortunate choice.

The first motivation to start our sensitivity simulations in the year 2002 is the statement given by Montzka et al. (2018) that there is "a gap in our understanding of CFC-11 sources and sinks since the early 2000s". As it is stated now clearer in our manuscript, our major goal is to create an upper limit

case (SEN-C2-fCFC11_2050), which should allow an assessment of maximum consequences. In addition, we have now included a second sensitivity simulation (SEN-C2-fCFC11_2020), which allows an estimation of maximum changes due to the enhanced CFC-11 emissions in the last 18 years (2002 – 2019).

We know from observations that atmospheric CFC-11 has further decreased since 2002, namely from about 258 ppt to about 230 ppt in 2017. That means that the sensitivity simulation assumes too high atmospheric CFC-11 for the time period 2002-2017. Thanks to its long lifetime this additional CFC-11 stays for a while in the atmosphere and makes a quantitative estimate of the recently discovered increased CFC-11 emissions for future ozone meaningless. I would have understood a sensitivity experiment that follows the observations until 2017 and assumes constant CFC-11 values afterwards.

As already said in our first reply to the referee, we understand the concerns of the referee regarding the set-up of our sensitivity simulation. Here we would like to explain our decision to start our sensitivity simulations in year 2002:

Taking the CFC-11 mean surface mixing ratio of the year 2002 and a constant value until 2050 was also motivated by Figure 2a in Montzka et al. (2018) and by Figure ES-2 in WMO (2018). Here they showed stable emissions between 2002 and 2012, but also they clearly indicated an increase of the emissions after 2012 (until 2017). In the meantime, the updated CFC-11 emissions shown by Harris et al. (2019; their Figure 1) indicated a further increase in 2018. We are aware that stable emissions are not equal to stable concentrations. To work with a constant background value is of course a rough (crude) assessment of the future evolution of CFC-11. But since the future evolution of CFC-11 emissions (and also surface mixing ratio) is uncertain, such a simplified assumption of a constant surface mixing ratio is from our point of view certainly justified as an extreme scenario dating back to the point when expected emissions and observations started to diverge, and in particular because of the lack of more precise information about the future evolution of CFC-11 emissions. This is now also stated in the manuscript (second part of the Introduction starting on page 2, line 27; and also see the second part of Section 2 on pages 4 and 5).

If we would follow the suggestion given by referee #1, we are convinced that most of our results will only change slightly quantitatively, but not qualitatively. The related investigation carried out here is focusing mostly on differences between the REF-C2 and SEN-C2-fCFC11_2050 simulations. To our understanding, starting the SEN-C2-fCFC11_2050 simulation in the year 2017 (instead of 2002) and running it until 2065 (instead of 2050) would lead to similar results. In Table 5A-3 (WMO, 2011) the mean surface CFC-11 mixing ratios (and used in our REF-C2 simulation) are given as follow:

- 2002: 258,3 ppt
- 2015: 227,6 ppt
- 2017: 221,8 ppt
- 2020: 213,0 ppt
- 2050: 127,2 ppt
- 2065:  94,4 ppt

In our presented work we have used the surface mixing ratio for the years 2002 and 2050, which means that we have reduced the CFC-11 surface value in this time period by about 50%, i.e. **131,1 ppt difference** between the REF-C2 and SEN-C2-fCFC11_2050 simulation **in 2050**.

Following the suggestion by the referee this would mean: The alternative sensitivity simulation would be started in 2017 (with a constant surface mixing ratio of 221,8 ppt, as calculated from Table 5A-3) and this simulation would be performed until the year 2065. This would result in a difference of the CFC-11 surface mixing ratio of **127,4 ppt in 2065**. It turned out that this CFC-11 difference value is

similar to the one, which was calculated before (i.e. a difference of 131,1 ppt in 2050 between SEN-C2-fCFC11_2050 and REF-C2). Therefore the calculated and presented differences of the TCO and PCO changes (and also the results of the ozone budget changes) in our investigation should be very similar to the results of the alternatively suggested simulation by the referee. To our understanding the calculated ozone changes are primarily affected by the prescribed CFC-11 differences (between REF and SEN) rather than from the CFC-11 background value, which is of course different (by 36,5 ppt at the surface) in 2002 and 2017.

In the revised version of the manuscript, we have pointed out more clearly that our model set-up can be taken as an upper limit estimate ("worst case") regarding the CFC-11 background condition. It is now mentioned and discussed in the Introduction and the Discussion and Conclusion sections.

In my opinion this study requires additional efforts before becoming acceptable for publication in ACP. Either the authors perform a new sensitivity study with a more meaningful set-up (not necessarily the one outlined above, if there are better ideas), or they have at least to provide an estimate of the overestimated increase in stratospheric chlorine loading due to fixed CFC-11 levels between 2002 and 2017 and the subsequent ozone loss in their current sensitivity run.

We very much hope that the explanations given before are convincing the referee. From our point of view such an alternative sensitivity study (as suggested by the reviewer, covering 2017-2065 with fixed concentrations after 2017) will not lead to qualitative changes in our results. Moreover, we have demonstrated (see also the next point of our reply to the referee's comment) that the required CFC-11 emissions for establishing a constant surface mixing ratio in our sensitivity simulation SEN-C2-fCFC11_2050 after 2002 (i.e. a mean of about 90 Gg/year) are significantly higher than observed, but that they are not exorbitantly higher than the observed emissions in the last years. Therefore our approach for an assessment of the upper limit of consequences is definitely justified.

In addition we have performed another sensitivity simulation (SEN-C2-fCFC11_2020), which allows a rough estimation of the (maximum) consequences related to the enhanced CFC-11 levels in the past 18 years (2002 – 2019) of our numerical exercise.

Specific comments by reviewer #1:
- No matter which constant CFC-11 value the authors assume for their sensitivity, it would be interesting to estimate the emissions required to achieve or maintain these CFC-11 values. This would help to put the made assumptions into perspective, also with historic CFC-11 emissions, and to get an idea of how likely the chosen scenario is.

The exact emissions, which are required to achieve the corresponding CFC-11 surface mixing ratio, have been calculated. The respective CFC-11 emission values are now presented and discussed in Section 2 (lower part of page 4 and page 5). The tracer nudging procedure applied in our CCM EMAC diagnoses the amount of CFC-11, which is necessary to adjust to the prescribed surface mixing ratio. The cumulative CFC-11 emissions in the REF-C2 simulation (from 2002 to 2050) result in about 400 Gg. In the sensitivity simulation SEN-C2-fCFC11_2050, where the CFC-11 mean surface mixing ratio is kept constant at $258.3 \times 10^{-12}$ mol/mol after the year 2002, the CFC-11 emissions required to achieve the constant surface mixing ratio value in our model after year 2002 is about 90 Gg/year (e.g. for year 2003 it is 87 Gg/year; the emissions in our model simulation are slightly increasing with time). The cumulative CFC-11 emissions (from 2002 to 2050) result in about 4500 Gg (i.e. roughly 4100 Gg more than in REF-C2). The emission values derived from observations given by Montzka et al. (2018)

are about 65 Gg/year (mean) for 2002 to 2012 and 75 Gg/year from 2014 to 2016 (see also Rigby et al., 2019). The figures presented by Rigby et al. (2019) and Harris et al. (2019, their Figure 1) with respect to the temporal evolution of CFC-11 emissions indicate a further increase after 2016 (incl. the 2018 value). In our second sensitivity simulation SEN-C2-fCFC11_2020 the cumulative CFC-11 emissions (from 2002 to 2050) result in about 2100 Gg (i.e. roughly 1700 Gg more than in REF-C2).

In summary, based on these calculated emission values derived from our model simulations it can be demonstrated that the prescribed surface boundary conditions are quite reasonable for an upper limit assessment of possible implications of enhanced CFC-11 levels.

- In general I would appreciate to see some information on the statistical significance of the displayed differences between both model simulations.

Thank you for this suggestion. We have prepared the desired information regarding the level of confidence of the long-term changes, i.e. the (multiple) linear regression and the corresponding uncertainties of the trend estimates (we use $\pm 2$ standard errors as rough equivalent of the 95% confidence interval). The corresponding values are presented in revised Section 3.2. We did not focus on the statistical significance of the results as we know the general processes behind the additional ozone loss, i.e. we know that we put in additional chlorine, which will change ozone abundances.

- As this study is based on one CCM only, it would be interesting to see a short discussion about the sensitivity of ozone recovery and return dates in EMAC to stratospheric chlorine compared to other CCMs, following the Dhomse et al. paper.

The CCM EMAC results with respect to REF-C2 are part of the Dhomse et al. (2019) paper. A short paragraph about the general quality of our EMAC results is provided at the end of the revised Introduction (page 3, line 25 and following), allowing a rating. Another classification of our model results is given in the final section (Discussions and Conclusion; page 11).

- Fig. 4, 5, and 6 show Antarctic ozone chemistry for September. Usually October is shown for Antarctic ozone. I assume the authors chose September because Fig. 2 shows the largest difference between both model simulations in September. Some explanation would be helpful.

We now added some additional explanations in the revised manuscript. Yes, we chose September because it shows the largest ozone changes. But also because September is less noisy compared to October (see e.g. Solomon et al., Science, 353, Issue 6296, 269-274, doi: 10.1126/science.aae0061, 2016). Two sentences have been added, see page 6, lines 24 to 26.

- p4, l15-17: What is the percentage increase of ClOx in the LS?

The percentage increase of $ClO_x$ in the US and LS for both sensitivity simulations are now presented in Section 3.1 (page 5, line 30 and the following lines).

- p6, l21-26: Do you attribute the discussed additional cooling in SEN-C2-fCFC11 to the additional CFC-11 or the changes in stratospheric ozone or both?

Regarding this point, a short statement is presented in Section 3.3 (page 9, lines 5 to 8) of the revised manuscript. It is difficult to determine the individual contributions of the additional cooling in a coupled CCM. We assume that both processes (i.e. radiative cooling by enhanced CFC-11 concentrations and by less ozone in the stratosphere caused by enhanced chlorine loading) will lead to the calculated additional cooling in the sensitivity simulation SEN-C2-fCFC11_2050.

Technical corrections:
- p2, l9: Dohmse -> Dhomse
- p8, l15: Dohmse -> Dhomse

Of course! Sorry. It has been changed.

- Fig. 1, 3, 4, 5: I think one running mean, either 3 or 5 years, would be enough. Especially in Fig. 3 (top) the many different lines are rather confusing than helpful.

All figures (now Figures 2, 4, 5, 6, and also the new Figure 7) are now only showing the running 5-year mean.

We think that the manuscript has been improved by considering the comments of this referee. We very much hope that the referee agrees that this work contains interesting results, which are helpful for the rating of consequences regarding the unexpected CFC-11 emission in recent years. From our point of view such a sensitivity study gives interesting results due to enhanced CFC-11 surface mixing ratios in recent and coming years, whereas other ODS are declining as expected. Beyond that, the detailed ozone budget analysis at the end of our study further provides some new and interesting results regarding compensation and buffering effects associated with the different production and loss cycles (see the discussion on page 10, bottom paragraph).

ACP-2019-239

Old title: **Implications of constant CFC-11 concentrations for the future ozone layer**

Revised title: **Possible implications of enhanced CFC-11 concentrations on ozone**

by Martin Dameris et al.

Reply to the referee #2 comments

We thank the referee for taking the time to read our manuscript and for the critical statements and comments. Although the reviewer has concerns about the scientific content and value of the first version of our manuscript, we have prepared a revised version containing significant changes, which are related to the points raised by this referee. Among others, another sensitivity simulation has been performed and the results are discussed. From our point of view the scientific content of our study has clearly improved and the general statement of our investigation is now much clearer. The title of the paper has been changed accordingly. We very much hope to convince the referee that this piece of work contains some interesting and new results, which should be published.

In the revised manuscript, we have now provided a much clearer explanation what the objectives of this study are. Apart from this it is our intention to make it clearer for what purpose this study should not be taken, i.e. that it is not a robust projection of future ozone changes under different conditions regarding CFC-11 (hence, to avoid confusion, also the title has been changed). In the meantime, our strategy regarding our CCM sensitivity simulations has received some support: The statements given by Harris et al. (2019; SPARC newsletter, summary of the CFC-11 workshop in Vienna, March 2019) are very clear in the sense that more concrete and more realistic projections of future CFC-11 levels are currently not possible. Therefore, our approach is justified, allowing a simplified ("worst case") assessment of ozone changes due to enhanced CFC-11. Our second sensitivity study, which is now included, has been performed to provide a clearer picture with respect to the question on how much the recent CFC-11 changes so far have already affected ozone and how important the consequences are. Again this estimate can be taken as an upper limit related to new emissions.

In the following we give our answers regarding all points raised by the reviewer. The statements and comments raised by the referee are printed in black and our replies are presented in blue. If possible, we are pointing to the corresponding revised text block or paragraph. The revised manuscript including all changes (highlighted) and a cleared version (without highlighted changes) are provided.

Answers to referee #2 and our explanations

Before we are commenting the expert's report on a point by point basis, we want to add a general explanation to set our work into context.

Our analysis presented here does not aim at specifically investigating the effect of the newly discovered emission and numerous possible "directly" related scenarios. The recent paper by Montzka et al. (2018) triggered our interest in investigating the effect of enhanced CFC-11concentrations. Our objective is to create an upper limit case (SEN-C2-fCFC11_2050), which should allow an assessment of maximum consequences. In particular we think that our numerical sensitivity study is definitely new and interesting because

- we create and analyze a specific case study where the surface CFC-11 mixing ratio is fixed for a specific time period whereas other ODS decline as expected;
- we provide (most likely) an upper limit for possible scenarios ("worst case"); this could be seen as "what should be avoided/what could happen as a worst case (e.g. if a lot of the new

production is actually stored in banks)" somewhat in the tradition of the Newman et al. (2009) paper;

- we investigate when and where ozone loss due to enhanced CFC-11 levels occurs;
- we perform a detailed ozone budget analysis determining for the first time in such a scenario compensation and buffering effects associated with different production and loss cycles;
- we show that even with these enhanced CFC-11 surface mixing ratio it takes time to see the effect on ozone (due to strong inter-annual variability); and
- we have performed another sensitivity simulation (SEN-C2-fCFC11_2020), which allows a rough estimation of the (maximum) consequences related to the enhanced CFC-11 levels in the past 18 years (2002 – 2019).

We very much hope that all these points are now much clearer described in the revised manuscript and that the message of our study is now more obvious.

Preface:

This paper details the effect of a very specific future scenario of CFC-11 on ozone recovery. It is a response to the recent measurements showing the CFC-11 values are not dropping as quickly as predicted by compliance to the Montreal Protocol and thus implying illegal emissions. It outlines this one simple scenario in sufficient detail and the paper is well written. However, it is severely lacking in answering the questions necessary to understand the effect of the newly discovered emissions (see below for details). Thus, I cannot recommend publication of this paper in anything like its present form. I suggest the authors rethink the scope of the problem and expand their study considerably.

The referee is right in saying that our model approach is simple. However, there is a big knowledge gap regarding CFC-11 emissions (see Harris et al., 2019). As said already, therefore our approach is justified, allowing a simplified ("worst case") assessment of ozone changes due to enhanced CFC-11. In the revised version of the manuscript, we have pointed out more clearly that our model set-up can be taken as an upper limit estimate. This is now mentioned and discussed in the Introduction and the Discussion and Conclusion sections.

We believe that on the basis of our model study we can answer relevant questions, which are related to the effects of the additional CFC-11 emissions (i.e. constant CFC-11 concentrations). From our point of view the sensitivity simulations (SEN-C2-fCFC11_2050 and SEN-C2-fCFC11_2020), which have been performed in addition to the reference simulation (REF-C2), are certainly suitable for an upper limit sensitivity study regarding CFC-11. Our investigation should not be considered as specific future scenario that we deem likely. It is not our aim to distill more specific scenarios. This study aims to assess the magnitude of possible ozone changes under constant CFC-11 surface mixing ratio. Since currently we do not have more detailed information about possible future CFC-11 levels, such a "simple" assessment is in our understanding justified. The goal of this paper is to answer questions, which are related to possible implications due to enhanced CFC-11 emissions, but we do not claim that this is the actual path we are on. Moreover, this paper was not intended to answer questions, which are focusing on the origin or sources of the additional CFC-11 emissions.

Furthermore, we do not aim at making a complete assessment, which currently makes no sense since the future evolution of CFC-11 emissions is uncertain. Our study rather aims to provide an insight into the reaction of the ozone layer with respect to enhanced CFC-11 levels in the coming years. (Further, once a new "most likely" emission scenario is agreed on we are willing to provide such a simulation. A "full" assessment would also mean that different CCMs performing the same scenario.)

Last but not least and already mentioned above, it is an interesting sensitivity study to keep CFC-11 constant, whereas other ODS are declining according to RCP scenarios; the results are partly new and

informative. To fully carve this out, we have for the first time performed a detailed ozone budget analysis for such sensitivity simulations.

General comments:

The recent paper of Montzka et al. 2018 makes a strong case that there are illegal emissions of CFC-11 presently occurring and that these have been occurring since 2012 and even perhaps earlier. This brings up many questions for future ozone recovery including (but not limited to):

With our model study, which is based on a comparison of two sensitivity simulations with a reference simulation using a chemistry-climate model (here EMAC), we would like to provide a qualitative and also a rough quantitative assessment of possible implications of enhanced CFC-11 levels. Among others, it should help to assess the order of magnitude of ozone changes, which can be expected due to enhanced CFC-11 emissions.

Have the emissions to date affected future ozone recovery;

To answer this question in more detail, we have performed a second sensitivity simulation (SEN-C2-fCFC11_2020) for the revised version of our manuscript. The results are presented and discussed, in particular in the context of Figure 7. It allows an estimation of maximum changes due to the enhanced CFC-11 emissions in the last 18 years (2002 – 2019). It is found that an immediate full implementation of the Montreal Protocol again (with a drop down to zero of the additional CFC-11 emissions after 2020) eventually leads back to the direction of the expected ozone conditions around the end of the 2040s, as they are calculated in our REF-C2 simulation. In the REF-C2 simulation the model-diagnosed CFC-11 emissions are nearly zero after about 2030, whereas in the SEN-C2-fCFC11_2020 simulations they are steadily decreasing from higher values in 2020 down to zero around year 2050 (see the last paragraph of Section 3.2, page 7).

How much more would continuation of the present emissions to various end dates affect ozone recovery;

Based on the results of our simulations, in particular based on SEN-C2-fCFC11_2050, we find a maximum delay of the closure of the ozone hole by about 20 years under these "extreme" conditions (see the paragraph at the end of the Discussions and Conclusion section). Our result is in line with other model estimates, which are mentioned by Harris et al. (2019). A first estimate presented in WMO (2018) showed that if total CFC-11 emissions were to continue at levels experienced from 2002–2016 (67 Gg/year), the return of mid-latitude and polar EESC to the 1980 value would be delayed by about 7 years and 20 years, respectively. In our SEN-C2-fCFC11_2050 simulation a mean annual CFC-11 emission of about 90 Gg is assumed (see the description in the second part of revised Section 2) and therefore our assessment can be used as upper limit estimation.

What if the emission increased, what would that do to future recovery;

We think that our SEN-C2-fCFC11_2050 simulation is acceptable as upper limit assessment of possible future consequences. Another more extreme scenario would certainly be possible, but from

our point of view such a study would be purely speculative. We know that the consequences are clearly depending on the future evolution of the CFC-11 emissions (see Harries et al., 2019). Instead, we have now performed the SEN-C2-fCFC11_2020 simulation and are discussing the results of it in the revised manuscript. Here we discuss the related model results, which are indicating that the consequences due to the enhanced CFC-11 emission in the last 18 years are not dramatic, if the international community would immediately react, i.e., if the CFC-11 (additional) emissions are stopped now (assuming that the banks have not considerably increased).

What if there are banks of CFC-11 (and perhaps CFC-12) associated with the illegal emission;

To answer this question is highly speculative because there are major uncertainties regarding other ODS, which are related to CFC-11 production (for instance CFC-12 and HCFC-22). As pointed out by Harris et al. (2019), there are "major uncertainties in quantifying the excess emissions to unreported production", which are due to "(i) a possible increase in the leakage from banks; and (ii) the influence of atmospheric variability". To our knowledge, so far there are no other obvious changes identified, which are related to the levels of other ODS than CFC-11. Nevertheless, it is clear that other additional emissions or influencing processes would have the potential to further impact the ozone layer. For our paper we have decided not to discuss the possible implications of other, enhanced ODS and other related changes.

The present paper does not address these questions in any detail. Instead it only addresses one simple scenario: if the mixing ratio of CFC-11 stays constant through 2050, what is the effect on ozone. This assumes that the emission rate of CFC-11 stays at a constant level slightly higher than any inferred emission estimated in Montzka et al. and that this emission stays constant for the next 3 decades.

We hope that based on our explanations given before most of the raised question by the reviewer are sufficiently answered. From our point of view all the mentioned points are now adequately considered and discussed in the revised version. Certainly many other scenarios seem to be possible and in principle they could be performed with our model (but this would be expensive regarding computational costs); the number of possible future CFC-11 scenarios is arbitrary high. And as it has been stated before, the range of uncertainties is very large, and therefore the discussion of results of the scenario simulations would contain a large part of speculation. From our point of view such a discussion would not be efficient.

The results presented in this study definitely help to rate the potential of enhanced CFC-11 levels. Certainly our investigation based on SEN-C2-fCFC11_2050 provides "only" a rough estimation of possible implications, but taking into account the large uncertainties regarding the circumstances with respect to the CFC-11 emissions itself and to the effects on other ODS, in our view such a simplified assumption of a constant surface mixing ratio is absolutely justified as a "worst case" dating back to the point when expected emissions and observations started to diverge (i.e. year 2002).

Ignoring that this scenario is unlikely to occur given the international response to this issue, the real problem with the paper is that so little of the problem space is explored. I see limited value in modeling one (unlikely) scenario in detail and ignoring all other possibilities.

Based on our extended explanations in the revised manuscript and in particular with respect to the presentation and discussion of the results of the additional sensitivity simulation, we hope to convince the referee that our study is useful and that the scientific value of our investigation is given. We definitely did not ignore other possibilities, but, as mentioned before, due to the fact of the large range of uncertainties, we think that such a simplified numerical approach should be allowed.

I can only assume this choice was made because it is easy to implement in their model and it only took one new run. Unfortunately, the only question answered is that if a larger emission than inferred in Montzka et al. is continued for three decades it will have a negative effect on ozone recovery.

For certain, our choice was definitely not made because it is the easiest way to implement such a sensitivity simulation (i.e. SEN-C2-fCFC11_2050) in our CCM EMAC. As previously said, we have a clear objective: provide an upper assessment of possible consequences due to enhanced CFC-11 levels; we try to investigate a worst case scenario. Our main interest is related to the question: "What must be avoided?" (to be similar to the paper by Newman et al. (2009) "World avoided" paper). However, we appreciate the remarks given by the referee, so this has eventually led to our second sensitivity simulation (SEN-C2-fCFC11_2020), which surely has improved our study.

This will surprise no one and in fact it can be predicted by computing the perturbation of EESC in 2050 by changing the CFC-11 mixing ratio between the ref value to the new value. This is a "back of the envelope" calculation.

We were very surprised by this statement of the referee, in particular in the light of the results presented in Section 3.3 (Stratospheric ozone budget). In this section, for the first time we have prepared and presented a detailed analysis of the model results with respect to the ozone production and loss cycles for such a scenario simulation, i.e. enhanced CFC-11 levels. There are some "surprises" (may be not only for us), for instance that the effects are smaller than expected due to some compensating effects by other ozone production and loss cycles. Our budget analysis based on REF-C2 and SEN-C2-fCFC11_2050 nicely shows (Figure 8) compensating effects for the global annual mean and in the upper stratosphere in the Southern Hemisphere polar region, whereas in the lower stratosphere in the Southern Hemisphere polar region the compensation does not occur. We show, which processes are buffering the additional ozone destruction through higher CFC-11 values. We show where (altitude) and at which time ozone is reduced compared to the REF-C2. Further, we show that the additional emissions need time to affect ozone. It would have taken years to register some deviation in the ozone columns due to the additional emissions. In addition, we have analyzed the effect of the temperature change on the chemistry (see discussion on page 9).

I expect much more of the problem space explored in a paper addressing the effect of illegal emissions of CFC-11 on ozone recovery and with a chemistry-climate model to use. As stated above, I recommend that the author team rethink the issues the Montzka et al. paper uncovered and take a real stab at helping answering them. It is necessary to frame the problem in terms of various possibilities for the emissions (and bank changes) and then from there predict mixing ratio scenarios and finally model time series. Only then can the reader understand the scope of the problem and the possible effects on future ozone.

A classification into different terms of various possibilities for the CFC-11 emissions and based on this predict mixing ratio scenarios is certainly beyond the scope of our study. Our study would take a different direction; it would raise completely different questions. We think that with this paper we are providing an important contribution to solve some of the problems of possible implications on enhanced CFC-11 levels. We hope that the reviewer can support our view and our scientific approach. In particular, we have rephrased the Introduction section and the Discussions and Conclusion section making the objectives of our numerical model study much clearer. A confrontation of the observed CFC-11 emissions (Montzka et al., 2018) with the emissions employed in our model simulations help to categorize our findings (see second part of Section 2). It is demonstrated that the corresponding CFC-11 emissions are reasonable in our model simulations, which are required to keep the surface mixing ratio constant. Therefore, the adopted simulations can be used as an upper estimate of possible consequences of enhanced CFC-11 emissions.

We very much hope that the referee can follow our argumentation line and that the reviewer agrees that the revised manuscript is acceptable for publication in ACP.

[revised manuscript text omitted]

---

## Author Response (AR2)

ACP 2019-0239

Possible implications of enhanced CFC-11 concentrations on ozone by M. Dameris et al.

Response to the referee comments on the revised version of our manuscript

Dear Editor, dear Jens-Uwe, all comments raised by referee #1 on our revised version of the paper are discussed and are adequately considered in the attached document – see below the version with track changes.

We hope that our replies and changes added in the manuscript are sufficient.

Thank you! Best regards,

Martin Dameris

**Possible implications of enhanced CFC-11 concentrations on ozone**

Martin Dameris, Patrick Jöckel, Matthias Nützel

Deutsches Zentrum für Luft- und Raumfahrt, Institut für Physik der Atmosphäre, Oberpfaffenhofen, Germany

*Correspondence to*: Martin Dameris (martin.dameris@dlr.de)

**Abstract.** This numerical model study is motivated by the observed global deviation from assumed emissions of chlorofluorocarbon-11 (CFC-11, $CFCl_3$) in recent years. Montzka et al. (2018) discussed a strong deviation of the assumed emissions of CFC-11 in the past 15 years, which indicates a violation of the Montreal Protocol for the protection of the ozone layer. An investigation is performed based on Chemistry-Climate Model (CCM) simulations that analyze the consequences of an enhanced CFC-11 surface mixing ratio. In comparison to a reference simulation (REF-C2), where a decrease of the CFC-11 surface mixing ratio of about 50% is assumed from the early 2000s to the middle of the century (i.e. mixing ratio in full compliance with the Montreal Protocol agreement), two sensitivity simulations are carried out: One simulation in which after the year 2002 the CFC-11 surface mixing ratio is kept constant until 2050 (SEN-C2-fCFC11_2050); this allows a qualitative estimate of possible consequences of high-level stable CFC-11 surface mixing ratio on the ozone layer ("worst case"). In the second sensitivity simulation, which is branched off from the first sensitivity simulation, it is assumed that starting in year 2020 the Montreal Protocol is fully implemented again, which leads to a delayed decrease of CFC-11 in this simulation (SEN-C2-fCFC11_2020) compared to the reference simulation; this enables a rough and most likely upper-limit assessment of how much the unexpected CFC-11 emissions to date have already affected ozone. In all three simulations climate evolves under the same greenhouse gas scenario (i.e. RCP 6.0) and all other ozone depleting substances are declining according to this scenario. Differences between the reference (REF-C2) and the two sensitivity simulations (SEN-C2-fCFC11_2050 and SEN-C2-fCFC11_2020) are discussed. In the SEN-C2-fCFC11_2050 simulation the total column ozone (TCO) in the 2040s (i.e. the years 2041-2050) is particularly affected in both polar regions in winter and spring. Maximum discrepancies of TCO are identified with reduced ozone values of up to around 30 Dobson Units in the Southern Hemisphere (SH) polar region during SH spring (in the order of 15%). An analysis of the respective partial column ozone (PCO) for the stratosphere indicates that strongest ozone changes are calculated for the polar lower stratosphere, where they are mainly driven by the enhanced stratospheric chlorine content and associated heterogeneous chemical processes. Furthermore, it turns out that the calculated ozone changes, especially in the upper stratosphere, are surprisingly small. For the first time in such a scenario we perform a complete ozone budget analysis regarding the production and loss cycles. The budget analysis shows that in the upper stratosphere the additional ozone depletion due to the catalysis by reactive chlorine is compensated partly by other processes related to enhanced ozone production or reduced ozone loss, for instance from nitrous oxide ($NO_x$). Based on the analysis of the SEN-C2-fCFC11_2020 simulation it turned

**Kommentar [DM1]:** Wording has been changed according to the referees comment. The words "worst case / worst case scenario" and "upper limit" has been avoided in the following. Corresponding text passages have been changed.

[revised manuscript text omitted]

**Kommentar [DM2]:** The reviewer is right with his/her statement regarding the point that the uncertainties of the CFC-11 sources and sinks. This has nothing to do with our motivation to start the sensitivity simulation in 2002. The text has been changed accordingly.

**Kommentar [DM3]:** This short paragraph has been added with respect to the first comment given by the referee („constant 2002 vs. 2017").

[revised manuscript text omitted]

**Kommentar [DM4]:** Wording has been changed according to the reviewer's comment.

**Kommentar [DM5]:** Thanks for the comment. Wording has been changed! It was not our aim to discuss here the recovery of the ozone layer.

[revised manuscript text omitted]

**Kommentar [DM6]:** An analysis as suggested by the referee is not trivial. Disentangling the various processes in a 3-d CCM requires additional diagnostics, as many of the involved processes are non-linearly connected with each other. The system is heading towards a different chemical equilibrium, because the distribution of educts and the temperature change.

To explain it a little bit, two sentences have been added in the manuscript. Based on our stored model output, a further detailed analysis is currently not possible. Please note that looking on specific tracers, such as ClONO2 does not really help to separate the temperature from the educts effect.

If the Editor thinks that additional analyses are necessary for this publication, we have to think about an adequate analysis technique.

[revised manuscript text omitted]

**Kommentar [DM9]:** The requested headings of the respective figures have been added.

**Ozone production rate**

[Figure]

Figure 8: The relative change of ozone production rates (in %), which are normalized to the total column production (through photolysis hυ, HO₂ and CH₃O₂) in the REF-C2 simulation. For the individual ozone destruction cycles and molecules mean difference values are shown, which have been derived from the REF-C2 and the SEN-C2-fCFC11_2050 (i.e. SEN minus REF) simulation for the 2040s (from 2041 to 2050). Left: for the mean annual global mean profiles; right: for the South polar region (70° S – 90° S) in September. Negative values are indicating an intensified ozone loss or a decreased ozone production in the SEN-C2-
fCFC11_2050 simulation, whereas higher values indicate more ozone production or less loss through a specific process. Thin horizontal lines indicate the nearest pressure levels to the model grid-boxes.